# Privately Learning Subspaces

**Vikrant Singhal**
Cheriton School of Computer Science
University of Waterloo
Waterloo, ON - N2L 3G1, Canada
`vikrant.singhal@uwaterloo.ca`

**Thomas Steinke**
Google Research, Brain Team
Mountain View, CA, United States of America
`subspace@thomas-steinke.net`

## Abstract

Private data analysis suffers a costly curse of dimensionality. However, the data often has an underlying low-dimensional structure. For example, when optimizing via gradient descent, the gradients often lie in or near a low-dimensional subspace. If that low-dimensional structure can be identified, then we can avoid paying (in terms of privacy or accuracy) for the high ambient dimension.

We present differentially private algorithms that take input data sampled from a low-dimensional linear subspace (possibly with a small amount of error) and output that subspace (or an approximation to it). These algorithms can serve as a pre-processing step for other procedures.

## 1 Introduction

Differentially private algorithms generally have a poor dependence on the dimensionality of their input. That is, their error or sample complexity grows polynomially with the dimension. For example, for the simple task of estimating the mean of a distribution supported on $[0,1]^d$, we have per-coordinate error $\Theta(\sqrt{d}/n)$ to attain differential privacy, where $n$ is the number of samples. In contrast, the non-private error is $\Theta(\sqrt{\log(d)/n})$.

This cost of dimensionality is inherent [BUV14, SU17, DSS+15]. *Any* method with lower error is susceptible to tracing attacks (a.k.a. membership inference attacks). However, these lower bounds only apply when the data distribution is "high-entropy." This leaves open the posssibility that we can circumvent the curse of dimensionality when the data has an underlying low-dimensional structure.

Data often does possess an underlying low-dimensional structure. For example, the gradients that arise in deep learning tend to be close to a low-dimensional subspace [ACG+16, LXT+17, GARD18, LFLY18, LGZ+20, ZWB20, FT20]. Low dimensionality can arise from meaningful relationships that are at least locally linear, such as income versus tax paid. It can also arise because we are looking at a function of data with relatively few attributes.

A long line of work [BLR08, HT10, HR10, Ull15, BBNS19, BCM+20, ZWB20, KRRT20, etc.] has shown how to exploit structure in the data to attain better privacy and accuracy. However, these approaches assume that this structure is known *a priori* or that it can be learned from non-private sources. This raises the question:

> Can we learn low-dimensional structure from the data subject to differential privacy?

We consider the simple setting where the data lies in $\mathbb{R}^d$ but is in, or very close to a linear subspace, of dimension $k$. We focus on the setting where $k \ll d$ and we develop algorithms whose sample

---

[0]The full version of this article is available online at `https://arxiv.org/abs/2106.00001`

35th Conference on Neural Information Processing Systems (NeurIPS 2021).

complexity does not depend on the ambient dimension $d$; a polynomial dependence on the true dimension $k$ is unavoidable.

Our algorithms identify the subspace in question or, if the data is perturbed slightly, an approximation to it. Identifying the subspace structure is interesting in its own right, but it also can be used as a pre-processing step for further analysis – by projecting to the low-dimensional subspace, we ensure subsequent data analysis steps do not need to deal with high-dimensional data.

## 1.1 Our Contributions: Privately Learning Subspaces – Exact Case

We first consider the exact case, where the data $X_1, \cdots, X_n \in \mathbb{R}^d$ are assumed to lie in a $k$-dimensional subspace (rather than merely being near to it) – i.e., $\mathsf{rank}\,(A) = k$, where $A = \sum_i^n X_i X_i^T \in \mathbb{R}^{d \times d}$. In this case, we can also recover the subspace exactly.

However, we must also make some non-degeneracy assumptions. We want to avoid a pathological input dataset such as the following. Suppose $X_1, \cdots, X_k$ are linearly independent, but $X_k = X_{k+1} = X_{k+2} = \cdots = X_n$. While we can easily reveal the repeated data point, we cannot reveal anything about the other points due to the privacy constraint.

A natural non-degeneracy assumption would be to assume that the data points are in "general position" – that is, that there are no non-trivial linear dependencies among the data points. This means that *every* set of $k$ data points spans the subspace or, equivalently, no subspace of dimension $k-1$ contains more than $k-1$ data points. This is a very natural assumption – if the data consists of $n$ samples from a continuous distribution on the subspace, then this holds with probability $1$. We relax this assumption slightly and assume that no subspace of dimension $k-1$ contains more than $\ell$ data points. We also assume that all points are non-zero. Note that we define subspaces to pass through the origin; our results can easily be extended to affine subspaces.

**Theorem 1.1** (Main Result – Exact Case). *For all $n, d, k, \ell \in \mathbb{N}$ and $\varepsilon, \delta > 0$ satisfying $n \geq O\left(\ell + \frac{\log(1/\delta)}{\varepsilon}\right)$, there exists a randomized algorithm $M : \mathbb{R}^{d \times n} \to \mathcal{S}_d^k$ satisfying the following. Here $\mathcal{S}_d^k$ denotes the set of all $k$-dimensional subspaces of $\mathbb{R}^d$.*

- *$M$ is $(\varepsilon, \delta)$-differentially private with respect to changing one column of its input.*

- *Let $X = (X_1, \cdots, X_n) \in \mathbb{R}^{d \times n}$. Suppose there exists a $k$-dimensional subspace $S_* \in \mathcal{S}_d^k$ that contains all but $\ell$ of the points – i.e., $|\{i \in [n] : X_i \in S_*\}| \geq n - \ell$. Further suppose that any $(k-1)$-dimensional subspace contains at most $\ell$ points – i.e., for all $S \in \mathcal{S}_d^{k-1}$, we have $|\{i \in [n] : X_i \in S\}| \leq \ell$. Then $\mathbb{P}\,[M(X) = S_*] = 1$.*

The parameter $\ell$ in Theorem 1.1 can be thought of as a robustness parameter. Ideally the data points are in general position, in which case $\ell = k - 1$. If a few points are corrupted, then we increase $\ell$ accordingly; our algorithm can tolerate the corruption of a small constant fraction of the data points. Theorem 1.1 is optimal in the sense that $n \geq \Omega\left(\ell + \frac{\log(1/\delta)}{\varepsilon}\right)$ samples are required.

## 1.2 Our Contributions: Privately Learning Subspaces – Approximate Case

Next we turn to the substantially more challenging approximate case, where the data $X_1, \cdots, X_n \in \mathbb{R}^d$ are assumed to be close to a $k$-dimensional subspace, but are not assumed to be contained within that subspace. Our algorithm for the exact case is robust to changing a few points, but very brittle if we change all the points by a little bit. Tiny perturbations of the data points (due to numerical errors or measurement imprecision) could push the point outside the subspace, which would cause the algorithm to fail. Thus it is important to for us to cover the approximate case and our algorithm for the approximate is entirely different from our algorithm for the exact case.

The approximate case requires us to precisely quantify how close the input data and our output are to the subspace and we also need to make quantitative non-degeneracy assumptions. It is easiest to formulate this via a distributional assumption. We will assume that the data comes from a Gaussian distribution where the covariance matrix has a certain eigenvalue gap. This is a strong assumption and we emphasize that this is only for ease of presentation; our algorithm works under weaker assumptions. Furthermore, we stress that the differential privacy guarantee is worst-case and does not depend on any distributional assumptions.

We assume that the data is drawn from a multivariate Gaussian $\mathcal{N}(0, \Sigma)$. Let $\lambda_1(\Sigma) \geq \lambda_2(\Sigma) \geq \cdots \geq \lambda_d(\Sigma)$ be the eigenvalues of $\Sigma \in \mathbb{R}^{d \times d}$. We assume that there are $k$ large eigenvalues $\lambda_1(\Sigma), \cdots, \lambda_k(\Sigma)$ – these represent the "signal" we want – and $d - k$ small eigenvalues $\lambda_{k+1}(\Sigma), \cdots, \lambda_d(\Sigma)$ – these are the "noise". Our goal is to recover the subspace spanned by the eigenvectors corresponding to the $k$ largest eigenvalues $\lambda_1(\Sigma), \cdots, \lambda_k(\Sigma)$. Our assumption is that there is a large *multiplicative* gap between the large and small eigenvalues. Namely, we assume $\frac{\lambda_{k+1}(\Sigma)}{\lambda_k(\Sigma)} \leq \frac{1}{\mathsf{poly}(d)}$.

**Theorem 1.2** (Main Result – Approximate Case). *For all $n, d, k \in \mathbb{N}$ and $\alpha, \gamma, \varepsilon, \delta > 0$ satisfying*

$$n \geq \Theta\left(\frac{k \log(1/\delta)}{\varepsilon} + \frac{\ln(1/\delta) \ln(\ln(1/\delta)/\varepsilon)}{\varepsilon}\right) \text{ and } \gamma^2 \leq \Theta\left(\frac{\varepsilon \alpha^2 n}{d^2 k \log(1/\delta)} \cdot \min\left\{\frac{1}{k}, \frac{1}{\log(k \log(1/\delta)/\varepsilon)}\right\}\right),$$

*there exists an algorithm $M : \mathbb{R}^{d \times n} \to \mathcal{S}_d^k$ satisfying the following. Here $\mathcal{S}_d^k$ is the set of all $k$-dimensional subspaces of $\mathbb{R}^d$ represented as projection matrices – i.e., $\mathcal{S}_d^k = \{\Pi \in \mathbb{R}^{d \times d} : \Pi^2 = \Pi = \Pi^T, \mathsf{rank}(\Pi) = k\}$.*

- *$M$ is $(\varepsilon, \delta)$-differentially private with respect to changing one column of its input.*

- *Let $X_1, \cdots, X_n$ be independent samples from $\mathcal{N}(0, \Sigma)$. Let $\lambda_1(\Sigma) \geq \lambda_2(\Sigma) \geq \cdots \geq \lambda_d(\Sigma)$ be the eigenvalues of $\Sigma \in \mathbb{R}^{d \times d}$. Suppose $\lambda_{k+1}(\Sigma) \leq \gamma^2 \cdot \lambda_k(\Sigma)$. Let $\Pi \in \mathcal{S}_d^k$ be the projection matrix onto the subspace spanned by the eigenvectors corresponding to the $k$ largest eigenvalues of $\Sigma$. Then $\mathbb{P}\left[\|M(X) - \Pi\| \leq \alpha\right] \geq 0.7$.*

The sample complexity of our algorithm $n = O(k \log(1/\delta)/\varepsilon)$ is independent of the ambient dimension $d$; this is ideal. However, there is a polynomial dependence on $d$ in $\gamma$, which controls the multiplicative eigenvalue gap. This multiplicative eigenvalue gap is a strong assumption, but it is also a necessary assumption if we want the sample complexity $n$ to be independent of the dimension $d$. In fact, it is necessary *even without the differential privacy constraint* [CZ16]. That is, if we did not assume an eigenvalue gap that depends polynomially on the ambient dimension $d$, then it would be impossible to estimate the subspace with sample complexity $n$ that is independent of the ambient dimension $d$ even in the non-private setting.

Our algorithm is based on the subsample and aggregate framework [NRS07] and a differentially private histogram algorithm. These methods are generally quite robust and thus our algorithm is, too. For example, our algorithm can tolerate $o(n/k)$ input points being corrupted arbitrarily. We also believe that our algorithm's utility guarantee is robust to relaxing the Gaussianity assumption. All that we require in the analysis is that the empirical covariance matrix of a few samples from the distribution is sufficiently close to its expectation $\Sigma$ with high probability.

## 1.3 Related Work

To the best of our knowledge, the problem of privately learning subspaces, as we formulate it, has not been studied before. However, a closely-related line of work is on Private Principal Component Analysis (PCA) and low-rank approximations. We briefly discuss this extensive line of work below, but first we note that, in our setting, all of these techniques have a sample complexity $n$ that grows polynomially with the ambient dimension $d$. Thus, they do not evade privacy's curse of dimensionality. However, we make a stronger assumption than these prior works – namely, we assume a large multiplicative eigenvalue gap. (Many of the prior works consider an *additive* eigenvalue gap, which is a weaker assumption.)

There has been a lot of interest in Private PCA, matrix completion, and low-rank approximation. One motivation for this is the infamous Netflix prize, which can be interpreted as a matrix completion problem. The competition was cancelled after researchers showed that the public training data revealed the private movie viewing histories of many of Netflix's customers [NS06]. Thus privacy is a real concern for matrix analysis tasks.

Many variants of these problems have been considered: Some provide approximations to the data matrix $X = (X_1, \cdots, X_n) \in \mathbb{R}^{d \times n}$; others approximate the covariance matrix $A = \sum_i^n X_i X_i^T \in \mathbb{R}^{d \times d}$ (as we do). There are also different forms of approximation – we can either produce a subspace or an approximation to the entire matrix, and the approximation can be measured by different norms (we consider the operator norm between projection matrices). Importantly, we define differential

privacy to allow one data point $X_i$ to be changed arbitrarily, whereas most of the prior work assumes a bound on the norm of the change or even assumes that only one coordinate of one vector can be changed. In the discussion below we focus on the techniques that have been considered for these problems, rather than the specific results and settings.

[DTTZ14] consider the simple algorithm which adds independent Gaussian noise to each of entries of the covariance matrix $A$, and then perform analysis on the noisy matrix. (In fact, this algorithm predates the development of differential privacy [BDMN05] and was also analyzed under differential privacy by McSherry and Mironov [MM09] and Chaudhuri, Sarwate, and Sinha [CSS12].) This simple algorithm is versatile and several bounds are provided for the accuracy of the noisy PCA. The downside of this is that a polynomial dependence on the ambient dimension $d$ is inherent – indeed, they prove a sample complexity lower bound of $n = \tilde{\Omega}(\sqrt{d})$ for any algorithm that identifies a useful approximation to the top eigenvector of $A$. This lower bound does not contradict our results because the relevant inputs do not satisfy our near low-rank assumption.

[HR12] and [ABU18] apply techniques from dimensionality reduction to privately compute a low-rank approximation to the input matrix $X$. [HR13] and [HP13] use the power iteration method with noise injected at each step to compute low-rank approximations to the input matrix $X$. In all of these, the underlying privacy mechanism is still noise addition and the results still require the sample complexity to grow polynomially with the ambient dimension to obtain interesting guarantees. (However, the results can be dimension-independent if we define differential privacy so that only one entry – as opposed to one column – of the matrix $X$ can be changed by 1. This is a significantly weaker privacy guarantee.)

[BBDS12] and [She19] also use tools from dimensionality reduction; they approximate the covariance matrix $A$. However, they show that the dimensionality reduction step itself provides a privacy guarantee (whereas the aforementioned results did not exploit this and relied on noise added at a later stage). [She19] analyzes two additional techniques – the addition of Wishart noise (i.e., $YY^T$ where the columns of $Y$ are independent multivariate Gaussians) and sampling from an inverse Wishart distribution (which has a Bayesian interpretation).

[CSS12], [KT13], [WSC$^+$16], and [ADK$^+$18] apply variants of the exponential mechanism [MT07] to privately select a low-rank approximation to the covariance matrix $A$. This method is nontrivial to implement and analyse, but it ultimately requires the sample complexity to grow polynomially in the ambient dimension.

[GGB18] exploit smooth sensitivity [NRS07] to release a low-rank approximation to the matrix $A$. This allows adding less noise than worst case sensitivity, under an eigenvalue gap assumption. However, the sample complexity $n$ is polynomial in the dimension $d$.

**Limitations of Prior Work**   Given the great variety of techniques and analyses that have been applied to differentially private matrix analysis problems, what is missing? We see that almost all of these techniques are ultimately based on some form of noise addition or the exponential mechanism. With the singular exception of the techniques of Sheffet [She19], all of these prior techniques satisfy pure[1] or concentrated differential privacy [BS16]. This is enough to conclude that these techniques cannot yield the dimension-independent guarantees that we seek. No amount of postprocessing or careful analysis can avoid this limitation. This is because pure and concentrated differential privacy have strong group privacy properties, which means "packing" lower bounds [HT10] apply.

We briefly sketch why concentrated differential privacy is incompatible with dimension-independent guarantees. Let the input be $X_1 = X_2 = \cdots = X_n = \xi/\sqrt{d}$ for a uniformly random $\xi \in \{-1, +1\}^d$. That is, the input is one random point repeated $n$ times. If $M$ satisfies $O(1)$-concentrated differential privacy, then it satisfies the mutual information bound $I(M(X); X) \leq O(n^2)$ [BS16]. But, if $M$ provides a meaningful approximation to $X$ or $A = XX^T$, then we must be able to recover an approximation to $\xi$ from its output, whence $I(M(X); X) \geq \Omega(d)$, as the entropy of $X$ is $d$ bits. This gives a lower bound of $n \geq \Omega(\sqrt{d})$, even though $X$ and $A$ have rank $k = 1$.

The above example shows that, even under the strongest assumptions (i.e., the data lies exactly in a rank-1 subspace), any good approximation to the subspace, to the data matrix $X$, or to the covariance matrix $A = XX^T$ must require the sample complexity $n$ to grow polynomially in the ambient

---

[1]Pure differential privacy (a.k.a. pointwise differential privacy) is $(\varepsilon, \delta)$-differential privacy with $\delta = 0$.

dimension $d$ if we restrict to techniques that satisfy concentrated differential privacy. Almost all of the prior work in this general area is subject to this restriction.

To avoid a sample complexity $n$ that grows polynomially with the ambient dimension $d$, we need fundamentally new techniques.

## 1.4 Our Techniques

For the exact case, we construct a score function for subspaces that has low sensitivity, assigns high score to the correct subspace, and assigns a low score to all other subspaces. Then we can simply apply a GAP-MAX algorithm to privately select the correct subspace [BDRS18].

The GAP-MAX algorithm satisfies $(\varepsilon, \delta)$-differential privacy and outputs the correct subspace as long as the gap between its score and that of any other subspace is larger than $O(\log(1/\delta)/\varepsilon)$. This works even though there are infinitely many subspaces to consider, which would not be possible under concentrated differential privacy.

The simplest score function would simply be the number of input points that the subspace contains. This assigns high score to the correct subspace, but it also assigns high score to any larger subspace that contains the correct subspace. To remedy this, we subtract from the score the number of points contained in a strictly smaller subspace. That is, the score of subspace $S$ is the number of points in $S$ minus the maximum over all subspaces $S' \subsetneq S$ of the number of points contained in $S'$.

This GAP-MAX approach easily solves the exact case, but it does not readily extend to the approximate case. If we count points near to the subspace, rather than in it, then (infinitely) many subspaces will have high score, which violates the assumptions needed for GAP-MAX to work. Thus we use a completely different approach for the approximate case.

We apply the "subsample and aggregate" paradigm of [NRS07]. That is, we split the dataset $X_1, \cdots, X_n$ into $n/O(k)$ sub-datasets each of size $O(k)$. We use each sub-dataset to compute an approximation to the subspace by doing a (non-private) PCA on the sub-dataset. Let $\Pi$ be the projection matrix onto the correct subspace and $\Pi_1, \cdots, \Pi_{n/O(k)}$ the projection matrices onto the approximations derived from the sub-datasets. With high probability $\|\Pi_j - \Pi\|$ is small for most $j$. (Exactly how small depends on the eigengap.) Now we must privately aggregate the projection matrices $\Pi_1, \cdots, \Pi_{n/O(k)}$ into a single projection matrix.

Rather than directly trying to aggregate the projection matrices, we pick a set of reference points, project them onto the subspaces, and then aggregate the projected points. We draw $p_1, \cdots, p_{O(k)}$ independently from a standard spherical Gaussian. Then $\|\Pi_j p_i - \Pi p_i\| \leq \|\Pi_j - \Pi\| \cdot O(\sqrt{k})$ is also small for all $i$ and most $j$. We wish to privately approximate $\Pi p_i$ and to do this we have $n/O(k)$ points $\Pi_j p_i$ most of which are close to $\Pi p_i$. This is now a location or mean estimation problem, which we can solve privately. Thus we obtain points $\hat{p}_i$ such that $\|\hat{p}_i - \Pi p_i\|$ is small for all $i$. From a PCA of these points we can obtain a projection $\hat{\Pi}$ with $\|\hat{\Pi} - \Pi\|$ being small, as required.

Finally, we discuss how to privately obtain $(\hat{p}_1, \hat{p}_2, \cdots, \hat{p}_{O(k)})$ from $(\Pi_1 p_1, \cdots, \Pi_1 p_{O(k)}), \cdots, (\Pi_{n/O(k)} p_1, \cdots, \Pi_{n/O(k)} p_{O(k)})$. It is better here to treat $(\hat{p}_1, \hat{p}_2, \cdots, \hat{p}_{O(k)})$ as a single vector in $\mathbb{R}^{O(kd)}$, rather than as $O(k)$ vectors in $\mathbb{R}^d$. We split $\mathbb{R}^{O(kd)}$ into cells and then run a differentially private histogram algorithm. If we construct the cells carefully, for most $j$ we have that $(\Pi_j p_1, \cdots, \Pi_j p_{O(k)})$ is in the same histogram cell as the desired point $(\Pi p_1, \cdots, \Pi p_{O(k)})$. The histogram algorithm will thus identify this cell, and we take an arbitrary point from this cell as our estimate $(\hat{p}_1, \hat{p}_2, \cdots, \hat{p}_{O(k)})$. The differentially private histogram algorithm is run over exponentially many cells, which is possible under $(\varepsilon, \delta)$-differential privacy if $n/O(k) \geq O(\log(1/\delta)/\varepsilon)$. (Note that under concentrated differential privacy the histogram algorithm's sample complexity $n$ would need to depend on the number of cells and, hence, the ambient dimension $d$.)

The main technical ingredients in the analysis of our algorithm for the approximate case are matrix perturbation and concentration analysis and the location estimation procedure using differentially private histograms. Our matrix perturbation analysis uses a variant of the Davis-Kahan theorem to show that if the empirical covariance matrix is close to the true covariance matrix, then the subspaces corresponding to the top $k$ eigenvalues of each are also close; this is applied to both the subsamples and the projection of the reference points. The matrix concentration results that we use show that the empirical covariance matrices in all the subsamples are close to the true covariance matrix. This is

the only place where the multivariate Gaussian assumption arises. Any distribution that concentrates well will work.

## 2 Exact case

Here, we discuss the case, where all $n$ points lie *exactly* in a subspace $s_*$ of dimension $k$ of $\mathbb{R}^d$. Our goal is to privately output that subspace. We do it under the assumption that all strict subspaces of $s_*$ contain at most $\ell$ points. If the points are in general position, then $\ell = k - 1$, as any strictly smaller subspace has dimension $< k$ and cannot contain more points than its dimension. Let $\mathcal{S}_d^k$ be the set of all $k$-dimensional subspaces of $\mathbb{R}^d$. Let $\mathcal{S}_d$ be the set of all subspaces of $\mathbb{R}^d$. We formally define that problem as follows.

**Problem 2.1.** *Assume (i) all but at most $\ell$, input points are in some $s_* \in \mathcal{S}_d^k$, and (ii) every subspace of dimension $< k$ contains at most $\ell$ points. (If the points are in general position – aside from being contained in $s_*$ – then $\ell = k - 1$.) The goal is to output a representation of $s_*$.*

We call these $\leq \ell$ points that do not lie in $s_*$, "adversarial points". With the problem defined in Problem 2.1, we will state the main theorem of this section.

**Theorem 2.2.** *For any $\varepsilon, \delta > 0$, $\ell \geq k - 1 \geq 0$, and*

$$n \geq O\left(\ell + \frac{\log(1/\delta)}{\varepsilon}\right),$$

*there exists an $(\varepsilon, \delta)$-DP algorithm $M : \mathbb{R}^{d \times n} \to \mathcal{S}_d^k$, such that if $X$ is a dataset of $n$ points satisfying the conditions in Problem 2.1, then $M(X)$ outputs a representation of $s_*$ with probability 1.*

We prove Theorem 2.2 by proving the privacy and the accuracy guarantees of Algorithm 1. The algorithm performs a GAP-MAX. It assigns a score to all the relevant subspaces, that is, the subspaces spanned by the points of the dataset $X$. We show that the only subspace that has a high score is the true subspace $s_*$, and the rest of the subspaces have low scores. Then GAP-MAX outputs the true subspace successfully because of the gap between the scores of the best subspace and the second to the best one. For GAP-MAX to work all the time, we define a default option in the output space that has a high score, which we call NULL. Thus, the output space is now $\mathcal{Y} = \mathcal{S}_d \cup \{\text{NULL}\}$. Also, for GAP-MAX to run in finite time, we filter $\mathcal{S}_d$ to select finite number of subspaces that have at least 0 scores on the basis of $X$. Note that this is a preprocessing step, and does not violate privacy as, we will show, all other subspaces already have 0 probability of getting output. We define the score function $u : \mathcal{X}^n \times \mathcal{Y} \to \mathbb{N}$ as follows.

$$u(x, s) := \begin{cases} |x \cap s| - \sup\{|x \cap t| : t \in \mathcal{S}_d, t \subsetneq s\} & \text{if } s \in \mathcal{S}_d \\ \ell + \frac{4\log(1/\delta)}{\varepsilon} + 1 & \text{if } s = \text{NULL} \end{cases}$$

Note that this score function can be computed in finite time because for any $m$ points and $i > 0$, if the points are contained in an $i$-dimensional subspace, then the subspace that contains all $m$ points must lie within the set of subspaces spanned by $\binom{m}{i+1}$ subsets of points.

### 2.1 Privacy

**Lemma 2.3.** *Algorithm 1 is $(\varepsilon, \delta)$-differentially private.*

The proof of Lemma 2.3 closely follows the privacy analysis of GAP-MAX by [BDRS18]. The only novelty is that Algorithm 1 may output NULL in the case that the input is malformed (i.e., doesn't satisfy the assumptions of Problem 2.1).

The key is that the score $u(X, s)$ is low sensitivity. Thus $\max\{0, u(X, s) - u(X, s_2) - 1\}$ also has low sensitivity. What we gain from subtracting the second-largest score and taking this maximum is that these values are also sparse – only one ($s = s_1$) is nonzero. This means we can add noise to all the values without paying for composition. We prove the privacy guarantees in the full version.

### 2.2 Accuracy

We start by showing that the true subspace $s_*$ has a high score, while the rest of the subspaces have low scores.

**Algorithm 1:** DP Exact Subspace Estimator $\text{DPESE}_{\varepsilon,\delta,k,\ell}(X)$

---

**Input:** Samples $X \in \mathbb{R}^{d \times n}$. Parameters $\varepsilon, \delta, k, \ell > 0$.
**Output:** $\hat{s} \in \mathcal{S}_d^k$.

1 Set $\mathcal{Y} \leftarrow \{\text{NULL}\}$ and sample noise $\xi(\text{NULL})$ from $\text{TLap}(2, \varepsilon, \delta)$.
2 Set score $u(X, \text{NULL}) = \ell + \frac{4\log(1/\delta)}{\varepsilon} + 1$.

   // Identify candidate outputs.
3 **for** *each subset $S$ of $X$ of size $k$* **do**
4     Let $s$ be the subspace spanned by $S$.
5     $\mathcal{Y} \leftarrow \mathcal{Y} \cup \{s\}$.
6     Sample noise $\xi(s)$ from $\text{TLap}(2, \varepsilon, \delta)$.
7     Set score $u(X, s) = |x \cap s| - \sup\{|x \cap t| : t \in \mathcal{S}_d, t \subsetneq s\}$.
8 **end**

   // Apply GAP-MAX.
9 Let $s_1 = \arg\max_{s \in \mathcal{Y}} u(X, s)$ be the candidate with the largest score.
10 Let $s_2 = \arg\max_{s \in \mathcal{Y} \setminus \{s_1\}} u(X, s)$ be the candidate with the second-largest score.
11 Let $\hat{s} = \arg\max_{s \in \mathcal{Y}} \max\{0, u(X, s) - u(X, s_2) - 1\} + \xi(s)$.
   // Truncated Laplace noise $\xi \sim \text{TLap}(2, \varepsilon, \delta)$

12 **return** $\hat{s}$.

---

**Lemma 2.4.** *Under the assumptions of Problem 2.1, $u(x, s_*) \geq n - 2\ell$ and $u(x, s') \leq 2\ell$ for $s' \neq s_*$.*

*Proof.* We have $u(x, s_*) = |x \cap s_*| - |x \cap s'|$ for some $s' \in \mathcal{S}_d$ with $s' \subsetneq s_*$. The dimension of $s'$ is at most $k - 1$ and, by the assumption (ii), $|x \cap s'| \leq \ell$.

Let $s' \in \mathcal{S}_d \setminus \{s_*\}$. There are three cases to analyse:

1. Let $s' \supsetneq s_*$. Then $u(x, s') \leq |x \cap s'| - |x \cap s_*| \leq \ell$ because the $\leq \ell$ adverserial points and the $\geq n - \ell$ non-adversarial points may not together lie in a subspace of dimension $k$.

2. Let $s' \subsetneq s_*$. Let $k'$ be the dimension of $s'$. Clearly $k' < k$. By our assumption (ii), $|s' \cap x| \leq \ell$. Then $u(x, s') = |x \cap s'| - |x \cap t| \leq \ell$ for some $t$ because the $\leq \ell$ adversarial points already don't lie in $s_*$, so they will not lie in any subspace of $s_*$.

3. Let $s'$ be incomparable to $s_*$. Let $s'' = s' \cap s_*$. Then $u(x, s') \leq |x \cap s'| - |x \cap s''| \leq \ell$ because the adversarial points may not lie in $s_*$, but could be in $s' \setminus s''$.

This completes the proof. $\qquad\square$

Now, we show that the algorithm is accurate.

**Lemma 2.5.** *If $n \geq 3\ell + \frac{8\log(1/\delta)}{\varepsilon} + 2$, then Algorithm 1 outputs $s_*$ for Problem 2.1.*

*Proof.* From Lemma 2.4, we know that $s_*$ has a score of at least $n - 2\ell$, and the next best subspace can have a score of at most $\ell$. Also, the score of $\text{NULL}$ is defined to be $\ell + \frac{4\log(1/\delta)}{\varepsilon} + 1$. This means that the gap satisfies $\max\{0, u(X, s_*) - u(X, s_2) - 1\} \geq n - 3\ell - \frac{4\log(1/\delta)}{\varepsilon} - 1$. Since the noise is bounded by $\frac{2\log(1/\delta)}{\varepsilon}$, our bound on $n$ implies that $\hat{s} = s_*$ $\qquad\square$

## 2.3 Extensions of our Algorithm

**Determining the True Dimension $k$.** Algorithm 1 actually does not need to know the parameter $k$, which is the dimension of the "true" subspace. That is, we could modify the algorithm to run over *all* subspaces, not just those of dimension $k$. Or, if we have an upper bound $k \leq k_{\max}$, then we could run the algorithm over all subspaces of dimension $\leq k_{\max}$. However, the analyst may want to know

what $k$ is before spending the privacy budget necessary to run the algorithm. If the dimension $k$ turns out to be large, then it may not be worth running the algorithm. In other words, it would be desirable to be able to compute the dimension $k$ without paying for the privacy cost of computing the subspace itself by running Algorithm 1.

We now sketch an algorithm that would help determine the right choice of $k$ when it is unkown, at a low cost in sample complexity. We essentially use the exponential mechanism [MT07] to find $k$.

Given $i = 1, \ldots, d$, let $S_i$ be the set of subspaces of dimension $i$ that could be found by taking all combinations of $i$ points from our dataset. Then we define $\text{SCORE}(i) := \max_{s \in S_i} u(X, s)$. Since $u$ has sensitivity 1, $\text{SCORE}(i)$ does too, as taking the maximum does not increase sensitivity. We then run the exponential mechanism over all $i$, and output the best $i$. This is private by the guarantees of the exponential mechanism. The intuition for accuracy is that only one subspace will have a high score (from the accuracy argument in the previous subsection), and all other subspaces will have low score. Therefore, the dimension of that subspace would be output with high probability by the mechanism, if the number of samples is at least $n \geq \frac{\bar{O}(\log(d))}{\varepsilon}$ since the sensitivity of $\text{SCORE}$ is 1. If we know an upper bound on $k$, say $k_{\max}$, we can evaluate over $i = 1, \ldots, k_{\max}$ and only pay for $k_{\max}$ instead of $d$.

Alternatively, if our goal is simply to determine whether $k \leq k_{\max}$ or $k > k_{\max}$, we could compute $\max_{i \leq k_{\max}} \text{SCORE}(i)$ and add Laplace or Gaussian noise. If this value is large, then probably $k \leq k_{\max}$, if it is small then probably $k > k_{\max}$. This would have minimal cost and may suffice to determine whether or not it is worth running the full algorithm.

**Computational Efficiency.** Algorithm 1 enumerates all subsets of input points of size $k$, so its running time is $O(n^k)$, which is not efficient unless $k$ is very small. We now sketch a modification to the algorithm that would make it efficient. Rather than considering all subsets we consider a random collection of subsets. That is, we change the `for` loop in Algorithm 1 so that instead of considering each subset $S$ of $X$ of size $k$ we consider some number of randomly chosen sets of $k$ points.

Intuitively, it suffices to consider such a random subset of the subspaces because any subspace $s$ with large $u(x, s)$ will be included in this subset with high probability. And any subspace $s$ with small $u(x, s)$ is not going to be output anyway. Formally, we must show that this new algorithm will with high probability output the same thing as the current algorithm. The failure probability is added to the $\delta$ of the $(\varepsilon, \delta)$-differential privacy guarantee. This requires changing the analysis as well as some of the parameter settings in the algorithm.

Note that our algorithm for the approximate case is computationally efficient. Thus, if computational efficiency is a concern, we can simply run that algorithm.

## 3 Approximate Case

In this section, we discuss the case, where the data "approximately" lies in a $k$-dimensional subspace of $\mathbb{R}^d$. We make a Gaussian distributional assumption, where the covariance is approximately $k$-dimensional, though the results could be extended to distributions with heavier tails using the right inequalities. We formally define the problem:

**Problem 3.1.** *Let $\Sigma \in \mathbb{R}^{d \times d}$ be a symmetric, PSD matrix of rank $\geq k \in \{1, \ldots, d\}$, and let $0 < \gamma \ll 1$, such that $\frac{\lambda_{k+1}}{\lambda_k} \leq \gamma^2$. Suppose $\Pi$ is the projection matrix corresponding to the subspace spanned by the eigenvectors of $\Sigma$ corresponding to the eigenvalues $\lambda_1, \ldots, \lambda_k$. Given sample access to $\mathcal{N}(\vec{0}, \Sigma)$, and $0 < \alpha < 1$, output a projection matrix $\widehat{\Pi}$, such that $\|\Pi - \widehat{\Pi}\| \leq \alpha$.*

We solve Problem 3.1 under the constraint of $(\varepsilon, \delta)$-differential privacy. Throughout this section, we would refer to the subspace spanned by the top $k$ eigenvectors of $\Sigma$ as the "true" or "actual" subspace.

Algorithm 2 solves Problem 3.1 and proves Theorem 1.2. Here $\| \cdot \|$ is the operator norm.

**Remark 3.2.** *We scale the eigenvalues of $\Sigma$ so that $\lambda_k = 1$ and $\lambda_{k+1} \leq \gamma^2$. Also, for the purpose of the analysis, we will be splitting $\Sigma = \Sigma_k + \Sigma_{d-k}$, where $\Sigma_k$ is the covariance matrix formed by the top $k$ eigenvalues and the corresponding eigenvectors of $\Sigma$ and $\Sigma_{d-k}$ is remainder. Also, we assume the knowledge of $\gamma$ and (an upper bound on) $\gamma$.*

Our solution is presented in Algorithm 2. The following theorem is the main result of the section.

**Theorem 3.3.** *Let* $\Sigma \in \mathbb{R}^{d \times d}$ *be an arbitrary, symmetric, PSD matrix of rank* $\geq k \in \{1, \ldots, d\}$, *and let* $0 < \gamma < 1$. *Suppose* $\Pi$ *is the projection matrix corresponding to the subspace spanned by the vectors of* $\Sigma_k$. *Then given*

$$\gamma^2 \in O\left(\frac{\varepsilon \alpha^2 n}{d^2 k \ln(1/\delta)} \cdot \min\left\{\frac{1}{k}, \frac{1}{\ln(k \ln(1/\delta)/\varepsilon)}\right\}\right),$$

*such that* $\lambda_{k+1}(\Sigma) \leq \gamma^2 \lambda_k(\Sigma)$, *for every* $\varepsilon, \delta > 0$, *and* $0 < \alpha < 1$, *there exists and* $(\varepsilon, \delta)$-*DP algorithm that takes*

$$n \geq O\left(\frac{k \log(1/\delta)}{\varepsilon} + \frac{\log(1/\delta) \log(\log(1/\delta)/\varepsilon)}{\varepsilon}\right)$$

*samples from* $\mathcal{N}(\vec{0}, \Sigma)$, *and outputs a projection matrix* $\widehat{\Pi}$, *such that* $\|\Pi - \widehat{\Pi}\| \leq \alpha$ *with probability at least* $0.7$.

Algorithm 2 is a type of "Subsample-and-Aggregate" algorithm [NRS07]. Here, we consider multiple subspaces formed by the points from the same Gaussian, and privately find a subspace that is close to all those subspaces. Since the subspaces formed by the points would be close to the true subspace, the privately found subspace would be close to the true subspace.

A little more formally, we first sample $q$ public data points (called "reference points") from $\mathcal{N}(\vec{0}, \mathbb{I})$. Next, we divide the original dataset $X$ into disjoint datasets of $m$ samples each, and project all reference points on the subspaces spanned by every subset. Now, for every reference point, we do the following. We have $t = \frac{n}{m}$ projections of the reference point. Using DP histogram over $\mathbb{R}^d$, we aggregate those projections in the histogram cells; with high probability all those projections will be close to one another, so they would lie within one histogram cell. We output a random point from the histogram cell corresponding to the reference point. With a total of $q$ points output in this way, we finally output the projection matrix spanned by these points. In the algorithm $C_0$, $C_1$, and $C_2$ are universal constants.

We divide the proof of Theorem 3.3 into two parts: privacy (Lemma 3.4) and accuracy.

## 3.1 Privacy

We analyse the privacy by understanding the sensitivities at the only sequence of steps invoking a differentially private mechanism, that is, the sequence of steps involving DP-histograms.

**Lemma 3.4.** *Algorithm 2 is* $(\varepsilon, \delta)$-*differentially private.*

*Proof.* Changing one point in $X$ can change only one of the $X^j$'s. This can only change one point in $Q$, which in turn can only change the counts in two histogram cells by 1. Therefore, the sensitivity is 2. Because the sensitivity of the histogram step is bounded by 2, an application of DP-histogram is $(\varepsilon, \delta)$-DP. Outputting a random point in the privately found histogram cell preserves privacy by post-processing. Hence, the claim. $\qquad\square$

## 3.2 Accuracy

The accuracy analysis of Algorithm 2 is relatively complex and is deferred to the full version. The key ingredients come from the literature on matrix concentration bounds and matrix perturbation inequalities. We briefly outline the key steps: First, we apply matrix concentration to show that the empirical covariance matrix $X^j(X^j)^T$ of each subsample is, after rescaling, close to the true covariance matrix $\Sigma$ with high probability. Second, we apply matrix perturbation inequalities to show that the top-$k$ subspace $\Pi_j$ corresponding to the data matrix $X^j$ is close to the true top-$k$ subspace $\Pi$. It follows that most of the the projected reference points $p_i^j$ are close to the desired value $\Pi p_i$. Third, we show that the aggregated projections $\hat{p}_i$ are also close to the true projections $\Pi_i$. Finally, we apply matrix perturbation inequalities again to show that the subspace derived from the aggregated projections $\widehat{\Pi}$ is close to the true subspace $\Pi$.

---

**Algorithm 2:** DP Approximate Subspace Estimator $\text{DPASE}_{\varepsilon,\delta,\alpha,\gamma,k}(X)$

---

**Input:** Samples $X_1, \ldots, X_n \in \mathbb{R}^d$. Parameters $\varepsilon, \delta, \alpha, \gamma, k > 0$.
**Output:** Projection matrix $\widehat{\Pi} \in \mathbb{R}^{d \times d}$ of rank $k$.

1   Set parameters: $t \leftarrow \frac{C_0 \ln(1/\delta)}{\varepsilon}$      $m \leftarrow \lfloor n/t \rfloor$      $q \leftarrow C_1 k$      $\ell \leftarrow \frac{C_2 \gamma \sqrt{dk}(\sqrt{k} + \sqrt{\ln(kt)})}{\sqrt{m}}$

2   Sample reference points $p_1, \ldots, p_q$ from $\mathcal{N}(\vec{0}, \mathbb{I})$ independently.

   `// Subsample from X, and form projection matrices.`
3   **for** $j \in 1, \ldots, t$ **do**
4      Let $X^j = (X_{(j-1)m+1}, \ldots, X_{jm}) \in \mathbb{R}^{d \times m}$.
5      Let $\Pi_j \in \mathbb{R}^{d \times d}$ be the projection matrix onto the subspace spanned by the eigenvectors of
       $X^j(X^j)^T \in \mathbb{R}^{d \times d}$ corresponding to the largest $k$ eigenvalues.
6      **for** $i \in 1, \ldots, q$ **do**
7        $p_i^j \leftarrow \Pi_j p_i$
8      **end**
9   **end**

   `// Create histogram cells with random offset.`
10   Let $\lambda$ be a random number in $[0, 1)$.
11   Divide $\mathbb{R}^{qd}$ into $\Omega = \{\ldots, [\lambda\ell + i\ell, \lambda\ell + (i+1)\ell), \ldots\}^{qd}$, for all $i \in \mathbb{Z}$.
12   Let each disjoint cell of length $\ell$ be a histogram bucket.

   `// Perform private aggregation of subspaces.`
13   For each $i \in [q]$, let $Q_i \in \mathbb{R}^{d \times t}$ be the dataset, where column $j$ is $p_i^j$.
14   Let $Q \in \mathbb{R}^{qd \times t}$ be the vertical concatenation of all $Q_i$'s in order.
15   Run $(\varepsilon, \delta)$-DP histogram over $\Omega$ using $Q$ to get $\omega \in \Omega$ that contains at least $\frac{t}{2}$ points.
16   **if** *no such $\omega$ exists* **then**
17      **return** $\perp$
18   **end**

   `// Return the subspace.`
19   Let $\widehat{p} = (\widehat{p}_1, \ldots, \widehat{p}_d, \ldots, \widehat{p}_{(q-1)d+1}, \ldots, \widehat{p}_{qd})$ be a random point in $\omega$.
20   **for** *each $i \in [q]$* **do**
21      Let $\widehat{p}_i = (\widehat{p}_{(i-1)d+1}, \ldots, \widehat{p}_{id})$.
22   **end**
23   Let $\widehat{\Pi}$ be the projection matrix of the top-$k$ subspace of $(\widehat{p}_1, \ldots, \widehat{p}_q)$.
24   **return** $\widehat{\Pi}$.

---

### 3.3   Handling Unknown Dimension $k$

Our algorithm requires knowning the parameter $k$, which determines the dimension of the subspace and which corresponds to the location of the eigenvalue gap. What if we only have an upper bound $k \leq k_{\max}$? In this case, we can use generic techniques to select the parameter $k$ [LT19, PS21]. These methods repeat our algorithm multiple times with different settings of the parameter $k$ and output the one with the highest self-reported score. The key is that these methods incur only a constant factor overhead in the parameter $\varepsilon$ despite repeating the algorithm a super-constant number of times. (They do, however, incur a $\text{poly}(k)$ factor overhead in the parameter $\delta$.) To apply these results we must modify our algorithm to also output a quality score for the subspace it has computed, such that if the right parameter $k$ is chosen we have high score and if the wrong parameter is chosen we have low score. To compute this score we take auxiliary samples (i.e., we add to the sample complexity and ensure that these samples are independent from the generated subspaces). Given the projection matrix $\widehat{\Pi}$ returned by our algorithm and an auxiliary sample $x$, we compute the projection $\widehat{\Pi}x$ and the orthogonal component $x - \widehat{\Pi}x$. Our success metric simply counts how many auxiliary samples satisfy $\|x - \widehat{\Pi}x\| \leq \widehat{\gamma}\|x\|$ for some threshold $\widehat{\gamma}$.

## Funding Disclosure

V.S. is currently at the University of Waterloo supported by an NSERC Discovery Grant, and was at Northeastern University supported by NSF grants CCF-1750640, CNS-1816028, and CNS-1916020. V.S. was an intern at IBM during part of this work. T.S. is currently employed by Google, but was employed by IBM during part of this work.

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
