# Privately Learning Subspaces

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

 matrix with eigenvalues $\lambda_1 \geq \lambda_2 \geq \cdots \geq \lambda_d \geq 0$. Fix $k \in [d]$ and let $0 < \gamma \ll 1$, be such that $\frac{\lambda_{k+1}}{\lambda_k} \leq \gamma^2$. Suppose $\Pi$ is the projection matrix onto the subspace spanned by the eigenvectors of $\Sigma$ corresponding to the eigenvalues $\lambda_1, \ldots, \lambda_k$. Given sample access to $\mathcal{N}(\vec{0}, \Sigma)$, and $0 < \alpha < 1$, output a projection matrix $\widehat{\Pi}$, such that $\|\Pi - \widehat{\Pi}\| \leq \alpha$.

We solve Problem 5.1 under the constraint of $(\varepsilon, \delta)$-differential privacy. Throughout this section, we would refer to the subspace spanned by the top $k$ eigenvectors of $\Sigma$ as the "true" or "actual" subspace.

Algorithm 2 solves Problem 5.1 and proves Theorem 1.2. Here $\| \cdot \|$ is the operator norm.

**Remark 5.2.** We scale the eigenvalues of $\Sigma$ so that $\lambda_k = 1$ and $\lambda_{k+1} \leq \gamma^2$. Also, for the purpose of the analysis, we will be splitting $\Sigma = \Sigma_k + \Sigma_{d-k}$, where $\Sigma_k$ is the covariance matrix formed by the top $k$ eigenvalues and the corresponding eigenvectors of $\Sigma$ and $\Sigma_{d-k}$ is remainder. We assume knowledge of $k$ and (an upper bound on ) $\gamma$.

Algorithm 2 is a type of "Subsample-and-Aggregate" algorithm [NRS07]. We consider multiple subspaces, each given by a disjoint subset of the input points which all come from the same multivariate Gaussian. Our algorithm privately finds a subspace that is close to most of those subspaces. By concentration, most of these subspaces will be close to the true subspace, and thus the privately-found subspace will also be close to the true subspace.

A little more formally, we first sample $q$ public data points (called "reference points") from $\mathcal{N}(\vec{0}, \mathbb{I})$. Next, we divide the original dataset $X$ into disjoint datasets of $m$ samples each, and perform PCA on each subset to identify the rank-$k$ subspace that best captures those samples. Then we project each of the reference points onto each of the subspaces. Now we have $t = \frac{n}{m}$ projections of each reference point, which we will privately aggregate into a single point. Finally, the aggregated points can be used to recover an approximation to the true subspace. To perform the aggregation, we use a DP histogram over a partition of $\mathbb{R}^d$. Specifically, we randomly partition $\mathbb{R}^d$ into cells such that, with high probability, most the projections will lie within one histogram cell. Thus we can privately identify that cell and output a random point from that histogram cell as the aggregated point.

## 5.1 Privacy

The privacy analysis of our method follows the template of the subsample-and-aggregate framework [NRS07] and our privacy guarantee directly follows from that of the DP histogram subroutine.

**Lemma 5.3.** *Algorithm 2 is $(\varepsilon, \delta)$-differentially private.*

*Proof.* Changing one point in $X$ can change only one of the $X^j$'s. This can only change one point in $Q$, which in turn can only change the counts in two histogram cells by 1. Therefore, the sensitivity is 2. Because the sensitivity of the histogram step is bounded by 2 (Lemma 5.3), an application of DP-histogram, by Lemma A.15, is $(\varepsilon, \delta)$-DP. Outputting a random point in the privately found histogram cell preserves privacy by post-processing (Lemma A.12). Hence, the claim. $\square$

## 5.2 Accuracy

The accuracy analysis of Algorithm 2 is relatively complex and is deferred to the full version. The key ingredients come from the literature on matrix concentration bounds and matrix perturbation inequalities. We briefly outline the key steps: First, we apply matrix concentration to show that the empirical covariance matrix $X^j(X^j)^T$ of each subsample is, after rescaling, close to the true covariance matrix $\Sigma$ with high probability. Second, we apply matrix perturbation inequalities to show that the top-$k$ subspace $\Pi_j$ corresponding to the empirical covariance matrix $X^j(X^j)^T$ is close to the true top-$k$ subspace $\Pi$. It follows that most of the the projected reference points $p_i^j$ are close to the desired value $\Pi p_i$. Third, we show that the aggregated projections $\hat{p}_i$ are also close to the true projections $\Pi_i$. Finally, we apply matrix perturbation inequalities again to show that the subspace derived from the aggregated projections $\widehat{\Pi}$ is close to the true subspace $\Pi$.

## 6 Conclusion, Discussion, & Limitations of Our Work

We provide algorithms for the problem of privately learning subspaces where the sample complexity does not depend on the ambient dimension. This is the first time such results have been given and,

---

**Algorithm 2:** DP Approximate Subspace Estimator $\text{DPASE}_{\varepsilon,\delta,\alpha,\gamma,k}(X)$

---

**Input:** Samples $X_1, \ldots, X_n \in \mathbb{R}^d$. Parameters $\varepsilon, \delta, \alpha, \gamma, k > 0$.
**Output:** Projection matrix $\widehat{\Pi} \in \mathbb{R}^{d \times d}$ of rank $k$.

Set parameters: $t \leftarrow \frac{C_0 \ln(1/\delta)}{\varepsilon}$ $\qquad m \leftarrow \lfloor n/t \rfloor$ $\qquad q \leftarrow C_1 k$ $\qquad \ell \leftarrow \frac{C_2 \gamma \sqrt{d} k (\sqrt{k} + \sqrt{\ln(kt)})}{\sqrt{m}}$

Sample reference points $p_1, \ldots, p_q$ from $\mathcal{N}(\vec{0}, \mathbb{I})$ independently.

`// Subsample from X, and form projection matrices.`
**For** $j \in 1, \ldots, t$
    Let $X^j = (X_{(j-1)m+1}, \ldots, X_{jm}) \in \mathbb{R}^{d \times m}$.
    Let $\Pi_j \in \mathbb{R}^{d \times d}$ be the projection matrix onto the subspace spanned by the eigenvectors of
    $X^j(X^j)^T \in \mathbb{R}^{d \times d}$ corresponding to the largest $k$ eigenvalues.
    **For** $i \in 1, \ldots, q$
        $p_i^j \leftarrow \Pi_j p_i$

`// Create histogram cells with random offset.`
Let $\lambda$ be a random number in $[0, 1)$.
Divide $\mathbb{R}^{qd}$ into $\Omega = \{\ldots, [\lambda\ell + i\ell, \lambda\ell + (i+1)\ell), \ldots\}^{qd}$, for all $i \in \mathbb{Z}$.
Let each disjoint cell of length $\ell$ be a histogram bucket.

`// Perform private aggregation of subspaces.`
For each $i \in [q]$, let $Q_i \in \mathbb{R}^{d \times t}$ be the dataset, where column $j$ is $p_i^j$.
Let $Q \in \mathbb{R}^{qd \times t}$ be the vertical concatenation of all $Q_i$'s in order.
Run $(\varepsilon, \delta)$-DP histogram over $\Omega$ using $Q$ to get $\omega \in \Omega$ that contains at least $\frac{t}{2}$ points.
**If** *no such $\omega$ exists*
    **Return** $\perp$

`// Return the subspace.`
Let $\widehat{p} = (\widehat{p}_1, \ldots, \widehat{p}_d, \ldots, \widehat{p}_{(q-1)d+1}, \ldots, \widehat{p}_{qd})$ be a random point in $\omega$.
**For** *each $i \in [q]$*
    Let $\widehat{p}_i = (\widehat{p}_{(i-1)d+1}, \ldots, \widehat{p}_{id}) \in \mathbb{R}^d$.
Let $\widehat{\Pi}$ be the projection matrix onto the subspace spanned by the eigenvectors corresponding to
  the $k$ largest eigenvalues of $\sum_{i=1}^q \widehat{p}_i \widehat{p}_i^T$.
**Return** $\widehat{\Pi}$.

---

as discussed in §2.1, prior work in the general area of private matrix analysis uses techniques that fundamentally cannot achieve sample complexity that is independent of the ambient dimension.

To achieve dimension-independent sample complexity, we must make strong assumptions about the data. Specifically, we must assume that the data points lie in or very near to a low-dimensional subspace. This is a limitation of our work. However, we emphasize that such assumptions are necessary to obtain dimension-independent sample complexity *even in the non-private setting* [CZ16].

We believe that the specific parameters in our results can be improved. We conjecture that the $\gamma^2$ parameter in Theorem 1.2 (which controls the eigenvalue gap) can be improved. Specifically, the exponent on the ambient dimension $d$ seems like it could be improved. (Although we know that it cannot be eliminated entirely.)

Our eigenvalue gap assumption could also be relaxed – rather than requiring a gap between $\lambda_k$ and $\lambda_{k+1}$, we could require a gap between $\lambda_k$ and $\lambda_{k+\ell}$. However, this would require changing other aspects of the problem formulation.

We hope that our work inspires further work. Generally, we believe that exploiting structure in the data to avoid privacy's curse of dimensionality is a fruitful and valuable research direction.

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

## Checklist

1. For all authors...

   (a) Do the main claims made in the abstract and introduction accurately reflect the paper's contributions and scope? [Yes]

   (b) Did you describe the limitations of your work? [Yes]

   (c) Did you discuss any potential negative societal impacts of your work? [N/A]

   (d) Have you read the ethics review guidelines and ensured that your paper conforms to them? [Yes]

2. If you are including theoretical results...

   (a) Did you state the full set of assumptions of all theoretical results? [Yes]

   (b) Did you include complete proofs of all theoretical results? [Yes]

3. If you ran experiments...

   (a) Did you include the code, data, and instructions needed to reproduce the main experimental results (either in the supplemental material or as a URL)? [N/A]

   (b) Did you specify all the training details (e.g., data splits, hyperparameters, how they were chosen)? [N/A]

   (c) Did you report error bars (e.g., with respect to the random seed after running experiments multiple times)? [N/A]

   (d) Did you include the total amount of compute and the type of resources used (e.g., type of GPUs, internal cluster, or cloud provider)? [N/A]

4. If you are using existing assets (e.g., code, data, models) or curating/releasing new assets...

   (a) If your work uses existing assets, did you cite the creators? [N/A]

   (b) Did you mention the license of the assets? [N/A]

   (c) Did you include any new assets either in the supplemental material or as a URL? [N/A]

(d) Did you discuss whether and how consent was obtained from people whose data you're using/curating? [N/A]

(e) Did you discuss whether the data you are using/curating contains personally identifiable information or offensive content? [N/A]

5. If you used crowdsourcing or conducted research with human subjects...

(a) Did you include the full text of instructions given to participants and screenshots, if applicable? [N/A]

(b) Did you describe any potential participant risks, with links to Institutional Review Board (IRB) approvals, if applicable? [N/A]

(c) Did you include the estimated hourly wage paid to participants and the total amount spent on participant compensation? [N/A]

 **Appendix**

 **A Notations, Definitions, and Background Results**

 **A.1 Linear Algebra and Probability Preliminaries**

542 Here, we mention a few key technical results that we will be using to prove the main theorem for
543 the approximate case. Throughout this document, we assume that the dimension $d$ is larger than
544 some absolute constant, and adopt the following notation: for a matrix $A$ of rank $r$, we use $s_1(A) \geq$
545 $\cdots \geq s_r(A)$ to denote the singular values of $A$ in decreasing order, and use $\lambda_1(A) \geq \cdots \geq \lambda_r(A)$ to
546 denote the eigenvalues of $A$ in decreasing order; let $s_{\min}(A)$ denote the least, non-zero singular value
547 of $A$. We omit the parentheses when the context is clear. We begin by stating two results about matrix
548 perturbation theory. The first result says that if two matrices are close to one another in operator
549 norm, then their corresponding singular values are also close to one another.

550 Define
$$\|M\| := \sup\{\|Mx\|_2 : x \in \mathbb{R}^d, \|x\|_2 \leq 1\}$$
551 to be the operator norm with respect to the Euclidean vector norm.

**Lemma A.1** (Singular Value Inequality). *Let $A, B \in \mathbb{R}^{d \times n}$ and let $r = \min\{d, n\}$. Then for $1 \leq i, j \leq r$,*
$$s_{i+j-1}(A + B) \leq s_i(A) + s_j(B).$$

552 The following result gives a lower bound on the least singular value of sum of two matrices.

**Lemma A.2** (Least Singular Value of Matrix Sum). *Let $A, B \in \mathbb{R}^{d \times n}$. Then*
$$s_{\min}(A + B) \geq s_{\min}(A) - \|B\|.$$

The next result bounds the angle between the subspaces spanned by two matrices that are close to one another. Let $X \in \mathbb{R}^{d \times n}$ have the following SVD.

$$X = [U \quad U_\perp] \cdot \begin{bmatrix} \Sigma_1 & 0 \\ 0 & \Sigma_2 \end{bmatrix} \cdot \begin{bmatrix} V^T \\ V_\perp^T \end{bmatrix}$$

In the above, $U, U_\perp$ are orthonormal matrices such that $U \in \mathbb{R}^{d \times r}$ and $U_\perp \in \mathbb{R}^{d \times (d-r)}$, $\Sigma_1, \Sigma_2$ are diagonal matrices, such that $\Sigma_1 \in \mathbb{R}^{r \times r}$ and $\Sigma_2 \in \mathbb{R}^{(d-r) \times (n-r)}$, and $V, V_\perp$ are orthonormal matrices, such that $V \in \mathbb{R}^{n \times r}$ and $V_\perp \in \mathbb{R}^{n \times (n-r)}$. Let $Z \in \mathbb{R}^{d \times n}$ be a perturbation matrix, and $\hat{X} = X + Z$, such that $\hat{X}$ has the following SVD.

$$\hat{X} = \begin{bmatrix} \hat{U} & \hat{U}_\perp \end{bmatrix} \cdot \begin{bmatrix} \hat{\Sigma}_1 & 0 \\ 0 & \hat{\Sigma}_2 \end{bmatrix} \cdot \begin{bmatrix} \hat{V}^T \\ \hat{V}_\perp^T \end{bmatrix}$$

553 In the above, $\hat{U}, \hat{U}_\perp, \hat{\Sigma}_1, \hat{\Sigma}_2, \hat{V}, \hat{V}_\perp$ have the same structures as $U, U_\perp, \Sigma_1, \Sigma_2, V, V_\perp$ respectively.
554 Let $Z_{21} = U_\perp U_\perp^T Z V V^T$ and $Z_{12} = U U^T Z V_\perp V_\perp^T$. Suppose $\sigma_1 \geq \cdots \geq \sigma_r \geq 0$ are the singular
555 values of $U^T \hat{U}$. Let $\Theta(U, \hat{U}) \in \mathbb{R}^{r \times r}$ be a diagonal matrix, such that $\Theta_{ii}(U, \hat{U}) = \cos^{-1}(\sigma_i)$.

**Lemma A.3** (Sin($\Theta$) Theorem [CZ16]). *Let $X, \hat{X}, Z, Z_{12}, Z_{21}$ be defined as above. Denote $\alpha = s_{\min}(U^T \hat{X} V)$ and $\beta = \|U_\perp^T \hat{X} V_\perp\|$. If $\alpha^2 > \beta^2 + \min\{\|Z_{12}\|^2, \|Z_{21}\|^2\}$, then we have the following.*
$$\|Sin(\Theta)(U, \hat{U})\| \leq \frac{\alpha\|Z_{21}\| + \beta\|Z_{12}\|}{\alpha^2 - \beta^2 - \min\{\|Z_{12}\|^2, \|Z_{21}\|^2\}}$$

556 The next result bounds $\|Sin(\Theta)(U, \hat{U})\|$ in terms of the distance between $UU^T$ and $\hat{U}\hat{U}^T$.

**Lemma A.4** (Property of $\|Sin(\Theta)\|$ [CZ16]). *Let $U, \hat{U} \in \mathbb{R}^{d \times r}$ be orthonormal matrices, and let $\Theta(U, \hat{U})$ be defined as above in terms of $\hat{U}, U$. Then we have the following.*
$$\|Sin(\Theta)(U, \hat{U})\| \leq \|\hat{U}\hat{U}^T - UU^T\| \leq 2\|Sin(\Theta)(U, \hat{U})\|$$

557 The next result bounds the singular values of a matrix, whose columns are independent vectors from a
558 mean zero, isotropic distribution in $\mathbb{R}^d$. We first define the sub-Gaussian norm of a random variable.

**Definition A.5.** Let $X$ be a sub-Gaussian random variable. The sub-Gaussian norm of $X$, denoted by $\|X\|_{\psi^2}$, is defined as,

$$\|X\|_{\psi^2} = \inf\{t > 0 : \mathbb{E}\left[\exp(X^2/t^2)\right] \leq 2\}.$$

**Lemma A.6** (Theorem 4.6.1 [Ver18])**.** *Let $A$ be an $n \times m$ matrix, whose columns $A_i$ are independent, mean zero, sub-Gaussian isotropic random vectors in $\mathbb{R}^n$. Then for any $t \geq 0$, we have*

$$\sqrt{m} - CK^2(\sqrt{n} + t) \leq s_n(A) \leq s_1(A) \leq \sqrt{m} + CK^2(\sqrt{n} + t)$$

559  *with probability at least $1 - 2\exp(-t^2)$. Here, $K = \max_i \|A\|_{\psi^2}$ (sub-Gaussian norm of A).*

560  In the above, $\|A\|_{\psi^2} \in O(1)$ if the distribution in question is $\mathcal{N}(\vec{0}, \mathbb{I})$. The following corollary
561  generalises the above result for arbitrary Gaussians.

**Corollary A.7.** *Let $A$ be an $n \times m$ matrix, whose columns $A_i$ are independent, random vectors in $\mathbb{R}^n$ from $\mathcal{N}(\vec{0}, \Sigma)$. Then for any $t \geq 0$, we have*

$$(\sqrt{m} - CK^2(\sqrt{n} + t))\sqrt{s_n(\Sigma)} \leq s_n(A) \leq (\sqrt{m} + CK^2(\sqrt{n} + t))\sqrt{s_n(\Sigma)}$$

*and*

$$s_1(A) \leq (\sqrt{m} + CK^2(\sqrt{n} + t))\sqrt{s_1(\Sigma)}$$

562  *with probability at least $1 - 2\exp(-t^2)$. Here, $K = \max_i \|A\|_{\psi^2}$ (sub-Gaussian norm of A).*

563  *Proof.* First, we prove the lower bound on $s_n(A)$. Note that $s_n(A) = \min_{\|x\|>0} \frac{\|Ax\|}{\|x\|}$, and that the
564  columns of $\Sigma^{-\frac{1}{2}} A$ are distributed as $\mathcal{N}(\vec{0}, \mathbb{I})$. Therefore, we have the following.

$$\begin{aligned}
\min_{\|x\|>0} \frac{\|Ax\|}{\|x\|} &= \min_{\|x\|>0} \frac{\|\Sigma^{\frac{1}{2}}\Sigma^{-\frac{1}{2}}Ax\|}{\|x\|} \\
&= \min_{\|x\|>0} \frac{\|\Sigma^{\frac{1}{2}}\Sigma^{-\frac{1}{2}}Ax\|}{\|\Sigma^{-\frac{1}{2}}Ax\|} \frac{\|\Sigma^{-\frac{1}{2}}Ax\|}{\|x\|} \\
&\geq \min_{\|x\|>0} \frac{\|\Sigma^{\frac{1}{2}}\Sigma^{-\frac{1}{2}}Ax\|}{\|\Sigma^{-\frac{1}{2}}Ax\|} \min_{\|x\|>0} \frac{\|\Sigma^{-\frac{1}{2}}Ax\|}{\|x\|} \\
&\geq \min_{\|y\|>0} \frac{\|\Sigma^{\frac{1}{2}}y\|}{\|y\|} \min_{\|x\|>0} \frac{\|\Sigma^{-\frac{1}{2}}Ax\|}{\|x\|} \\
&\geq (\sqrt{m} - CK^2(\sqrt{n} + t))\sqrt{s_n(\Sigma)} \qquad \text{(Lemma A.6)}
\end{aligned}$$

565  Next, we prove the upper bound on $s_n(A)$. For this, we first show that for $X \in \mathbb{R}^{m \times d}$ and $Y \in \mathbb{R}^{d \times n}$,
566  $s_{\min}(XY) \leq s_{\min}(X) \cdot \|Y\|$.

$$\begin{aligned}
s_{\min}(XY) &= \min_{\|z\|=1} \|XYz\| \\
&\leq \min_{\|z\|=1} \|X\|\|Yz\| \\
&= \|X\| \cdot \min_{\|z\|=1} \|Yz\| \\
&= \|X\| \cdot s_{\min}(Y)
\end{aligned}$$

567  Now, $s_{\min}(XY) = s_{\min}(Y^T X^T) \leq \|Y\| \cdot s_{\min}(X)$ by the above reasoning. Using this results, we
568  have the following.

$$\begin{aligned}
s_n(A) &= s_n(\Sigma^{1/2} \cdot \Sigma^{-1/2}A) \\
&\leq s_n(\Sigma^{1/2})\|\Sigma^{-1/2}A\| \\
&\leq (\sqrt{m} + CK^2(\sqrt{n} + t))\sqrt{s_n(\Sigma)} \qquad \text{(Lemma A.6)}
\end{aligned}$$

569  Now, we show the upper bound on $s_1(A)$. Note that $s_1(A) = \|A\|$.

$$\begin{aligned}
\|A\| &= \|\Sigma^{\frac{1}{2}}\Sigma^{-\frac{1}{2}}A\| \\
&\leq \|\Sigma^{\frac{1}{2}}\| \cdot \|\Sigma^{-\frac{1}{2}}A\| \\
&\leq (\sqrt{m} + CK^2(\sqrt{n} + t))\sqrt{s_1(\Sigma)} \qquad \text{(Lemma A.6)}
\end{aligned}$$

570  This completes the proof. $\qquad\qquad\square$

571   Now, we state a concentration inequality for $\chi^2$ random variables.

**Lemma A.8.** *Let $X$ be a $\chi^2$ random variable with $k$ degrees of freedom. Then,*

$$\mathbb{P}\left[X > k + 2\sqrt{kt} + 2t\right] \leq e^{-t}.$$

572   Next, we state the well-known Bernstein's inequality for sums of independent Bernoulli random
573   variables.

**Lemma A.9** (Bernstein's Inequality). *Let $X_1, \ldots, X_m$ be independent Bernoulli random variables taking values in $\{0, 1\}$. Let $p = \mathbb{E}[X_i]$. Then for $m \geq \frac{5p}{2\varepsilon^2} \ln(2/\beta)$ and $\varepsilon \leq p/4$,*

$$\mathbb{P}\left[\left|\frac{1}{m}\sum X_i - p\right| \geq \varepsilon\right] \leq 2e^{-\varepsilon^2 m/2(p+\varepsilon)} \leq \beta.$$

574   We finally state a result about the norm of a vector sampled from $\mathcal{N}(\vec{0}, \mathbb{I})$.

**Lemma A.10.** *Let $X_1, \ldots, X_q \sim \mathcal{N}(\vec{0}, \Sigma)$ be vectors in $\mathbb{R}^d$, where $\Sigma$ is the projection of $\mathbb{I}_{d \times d}$ on to a subspace of $\mathbb{R}^d$ of rank $k$. Then*

$$\mathbb{P}\left[\forall i, \|X_i\|^2 \leq k + 2\sqrt{kt} + 2t\right] \geq 1 - qe^{-t}.$$

575   *Proof.* Since $\Sigma$ is of rank $k$, we can directly use Lemma A.8 for a fixed $i \in [q]$, and the union bound
576   over all $i \in [q]$ to get the required result. This is because for any $i$, $\|X_i\|^2$ is a $\chi^2$ random variable
577   with $k$ degrees of freedom. □

## A.2   Privacy Preliminaries

**Definition A.11** (Differential Privacy (DP) [DMNS06]). A randomized algorithm $M : \mathcal{X}^n \to \mathcal{Y}$ satisfies $(\varepsilon, \delta)$-differential privacy ($(\varepsilon, \delta)$-DP) if for every pair of neighboring datasets $X, X' \in \mathcal{X}^n$ (i.e., datasets that differ in exactly one entry),

$$\forall Y \subseteq \mathcal{Y} \quad \mathbb{P}\left[M(X) \in Y\right] \leq e^\varepsilon \cdot \mathbb{P}\left[M(X') \in Y\right] + \delta.$$

579   When $\delta = 0$, we say that $M$ satisfies $\varepsilon$-differential privacy or pure differential privacy.

580   Neighbouring datasets are those that differ by the replacement of one individual's data. In our setting,
581   each individual's data is assumed to correspond to one point in $\mathcal{X} = \mathbb{R}^d$, so neighbouring means one
582   point is changed arbitrarily.

583   Throughout the document, we will assume that $\varepsilon$ is smaller than some absolute constant less than
584   1 for notational convenience, but note that our results still hold for general $\varepsilon$. Now, this privacy
585   definition is closed under post-processing.

**Lemma A.12** (Post Processing [DMNS06]). *If $M : \mathcal{X}^n \to \mathcal{Y}$ is $(\varepsilon, \delta)$-DP, and $P : \mathcal{Y} \to \mathcal{Z}$ is any randomized function, then the algorithm $P \circ M$ is $(\varepsilon, \delta)$-DP.*

## A.3   Basic Differentially Private Mechanisms.

589   We first state standard results on achieving privacy via noise addition proportional to sensitiv-
590   ity [DMNS06].

**Definition A.13** (Sensitivity). Let $f : \mathcal{X}^n \to \mathbb{R}^d$ be a function, its $\ell_1$-*sensitivity* and $\ell_2$-*sensitivity* are

$$\Delta_{f,1} = \max_{X \sim X' \in \mathcal{X}^n} \|f(X) - f(X')\|_1 \quad \text{and} \quad \Delta_{f,2} = \max_{X \sim X' \in \mathcal{X}^n} \|f(X) - f(X')\|_2,$$

591   respectively. Here, $X \sim X'$ denotes that $X$ and $X'$ are neighboring datasets (i.e., those that differ in
592   exactly one entry).

593   One way of introducing $(\varepsilon, \delta)$-differential privacy is via adding noise sampled from the truncated
594   Laplace distribution, proportional to the $\ell_1$ sensitivity.

**Lemma A.14** (Truncated Laplace Mechanism [GDGK20]). *Define the probability density function (p) of the truncated Laplace distribution as follows.*

$$p(x) = \begin{cases} Be^{-\frac{|x|}{\lambda}} & \text{if } x \in [-A, A] \\ 0 & \text{otherwise} \end{cases}$$

*In the above,*

$$\lambda = \frac{\Delta}{\varepsilon}, \quad A = \frac{\Delta}{\varepsilon} \log\left(1 + \frac{e^\varepsilon - 1}{2\delta}\right), \quad B = \frac{1}{2\lambda(1 - e^{-\frac{A}{\lambda}})}.$$

*Let* $\text{TLap}(\Delta, \varepsilon, \delta)$ *denote a draw from the above distribution.*

*Let* $f : \mathcal{X}^n \to \mathbb{R}^d$ *be a function with sensitivity* $\Delta$. *Then the truncated Laplace mechanism*

$$M(X) = f(X) + \text{TLap}(\Delta, \varepsilon, \delta)$$

*satisfies* $(\varepsilon, \delta)$-*DP.*

In the above $A \leq \frac{\Delta_{f,1}}{\varepsilon} \log(1/\delta)$ since $\varepsilon$ is smaller than some absolute constant less than 1. Now, we introduce differentially private histograms.

**Lemma A.15** (Private Histograms). *Let* $n \in \mathbb{N}$, $\varepsilon, \delta, \beta > 0$, *and* $\mathcal{X}$ *a set. There exists* $M : \mathcal{X}^n \to \mathbb{R}^{\mathcal{X}}$ *which is* $(\varepsilon, \delta)$-*differentially private and, for all* $x \in \mathcal{X}^n$, *we have*

$$\mathbb{P}_M \left[ \sup_{y \in \mathcal{X}} \left| M(x)_y - \frac{1}{n} |\{i \in [n] : x_i = y\}| \right| \leq O\left( \frac{\log(1/\delta\beta)}{\varepsilon n} \right) \right] \geq 1 - \beta.$$

The above holds due to [BNS16; Vad17]. Finally, we introduce the GAP-MAX algorithm from [BDRS18] that outputs the element from the output space that has the highest score function, given that there is a significant gap between the scores of the highest and the second to the highest elements.

**Lemma A.16** (GAP-MAX Algorithm [BDRS18]). *Let* $\text{SCORE} : \mathcal{X}^n \times \mathcal{Y} \to \mathbb{R}$ *be a score function with sensitivity 1 in its first argument, and let* $\varepsilon, \delta > 0$. *Then there exists a* $(\varepsilon, \delta)$-*differentially private algorithm* $M : \mathcal{X}^n \to \mathcal{Y}$ *and* $\alpha = \Theta(\log(1/\delta)/\varepsilon n)$ *with the following property. Fix an input* $X \in \mathcal{X}^n$. *Let*

$$y^* = \arg\max_{y \in \mathcal{Y}} \{\text{SCORE}(X, y)\}.$$

*Suppose*

$$\forall y \in \mathcal{Y}, y \neq y^* \implies \text{SCORE}(X, y) < \text{SCORE}(X, y^*) - \alpha n.$$

*Then* $M$ *outputs* $y^*$ *with probability 1.*

# B  Proof of Privacy of Algorithm 1

*Proof of Lemma 4.2.* First, we argue that the sensitivity of $u$ is 1. The quantity $|X \cap s|$ has sensitivity 1 and so does $\sup\{|X \cap t| : t \in \mathcal{S}_d, t \subsetneq s\}$. This implies sensitivity 2 by the triangle inequality. However, we see that it is not possible to change one point that simultaneously increases $|X \cap s|$ and decreases $\sup\{|X \cap t| : t \in \mathcal{S}_d, t \subsetneq s\}$ or vice versa. Thus the sensitivity is actually 1.

We also argue that $u(X, s_2)$ has sensitivity 1, where $s_2$ is the candidate with the second-largest score. Observe that the second-largest score is a monotone function of the collection of all scores – i.e., increasing scores cannot decrease the second-largest score and vice versa. Changing one input point can at most increase all the scores by 1, which would only increase the second-largest score by 1.

This implies that $\max\{0, u(X, s) - u(X, s_2) - 1\}$ has sensitivity 2 by the triangle inequality and the fact that the maximum does not increase the sensitivity.

Now we observe that for any input $X$ there is at most one $s$ such that $\max\{0, u(X, s) - u(X, s_2) - 1\} \neq 0$, namely $s = s_1$. We can say something even stronger: Let $X$ and $X'$ be neighbouring datasets with $s_1$ and $s_2$ the largest and second-largest scores on $X$ and $s_1'$ and $s_2'$ the largest and second-largest scores on $X'$. Then there is at most one $s$ such that $\max\{0, u(X, s) - u(X, s_2) - 1\} \neq 0$ or $\max\{0, u(X', s) - u(X', s_2') - 1\} \neq 0$. In other words, we cannot have both $u(X, s_1) - u(X, s_2) > 1$ and $u(X', s_1') - u(X', s_2') > 1$ unless $s_1 = s_1'$. This holds because $u(X, s) - u(X, s_2)$ has sensitivity 2.

With these observations in hand, we can delve into the privacy analysis. Let $X$ and $X'$ be neighbouring datasets with $s_1$ and $s_2$ the largest and second-largest scores on $X$ and $s'_1$ and $s'_2$ the largest and second-largest scores on $X'$. Let $\mathcal{Y}$ be the set of candidates from $X$ and let $\mathcal{Y}'$ be the set of candidates from $X'$. Let $\breve{\mathcal{Y}} = \mathcal{Y} \cup \mathcal{Y}'$ and $\hat{\mathcal{Y}} = \mathcal{Y} \cap \mathcal{Y}'$.

We note that, for $s \in \breve{\mathcal{Y}}$, if $u(X, s) \leq \ell$, then there is no way that $\hat{s} = s$. This is because $|\xi(s)| \leq \frac{2 \log(1/\delta)}{\varepsilon}$ for all $s$ and hence, there is no way we could have $\arg\max_{s \in \mathcal{Y}} \max\{0, u(X, s) - u(X, s_2) - 1\} + \xi(s) \geq \arg\max_{s \in \mathcal{Y}} \max\{0, u(X, \mathsf{NULL}) - u(X, s_2) - 1\} + \xi(\mathsf{NULL})$.

If $s \in \breve{\mathcal{Y}} \setminus \hat{\mathcal{Y}}$, then $u(X, s) \leq |X \cap s| \leq k + 1 \leq \ell$ and $u(X', s) \leq \ell$. This is because $s \notin \hat{\mathcal{Y}}$ implies $|X \cap s| < k$ or $|X' \cap s| < k$, but $|X \cap s| \leq |X' \cap s| + 1$. Thus, there is no way these points are output and, hence, we can ignore these points in the privacy analysis. (This is the reason for adding the $\mathsf{NULL}$ candidate.)

Now we argue that the entire collection of noisy values $\max\{0, u(X, s) - u(X, s_2) - 1\} + \xi(s)$ for $s \in \hat{\mathcal{Y}}$ is differentially private. This is because we are adding noise to a vector where (i) on the neighbouring datasets only 1 coordinate is potentially different and (ii) this coordinate has sensitivity 2. $\qquad\square$

# C   Lower Bound for Exact Case

Here, we show that our upper bound is optimal up to constants for the exact case.

**Theorem C.1.** *Any $(\varepsilon, \delta)$-DP algorithm that takes a dataset of $n$ points satisfying the conditions in Problem 4.1 and outputs $s_*$ with probability $> 0.5$ requires $n \geq \Omega\left(\ell + \frac{\log(1/\delta)}{\varepsilon}\right)$.*

*Proof.* First, $n \geq \ell + k$. This is because we need at least $k$ points to span the subspace, and $\ell$ points could be corrupted. Second, $n \geq \Omega(\log(1/\delta)/\varepsilon)$ by group privacy. Otherwise, the algorithm is $(10, 0.1)$-differentially private with respect to changing the *entire* dataset and it is clearly impossible to output the subspace under this condition. $\qquad\square$

# D   Proof of Accuracy of Algorithm 2

Now we delve into the utility analysis of the algorithm. For $1 \leq j \leq t$, let $X^j$ be the subsets of $X$ as defined in Algorithm 2, and $\Pi_j$ be the projection matrices of their respective subspaces. We now show that $\Pi_j$ and the projection matrix of the subspace spanned by $\Sigma_k$ are close in operator norm.

**Lemma D.1.** *Let $\Pi$ be the projection matrix of the subspace spanned by the vectors of $\Sigma_k$, and for each $1 \leq j \leq t$, let $\Pi_j$ be the projection matrix as defined in Algorithm 2. If $m \geq O(k + \ln(qt))$, then*

$$\mathbb{P}\left[\forall j, \|\Pi - \Pi_j\| \leq O\left(\frac{\gamma\sqrt{d}}{\sqrt{m}}\right)\right] \geq 0.95$$

*Proof.* We show that the subspaces spanned by $X^j$ and the true subspace spanned by $\Sigma$ are close. Formally, we invoke Lemmata A.3 and A.4. This closeness follows from standard matrix concentration inequalities.

Fix a $j \in [t]$. Note that $X^j$ can be written as $Y^j + H$, where $Y^j$ is the matrix of vectors distributed as $\mathcal{N}(\vec{0}, \Sigma_k)$, and $H$ is a matrix of vectors distributed as $\mathcal{N}(\vec{0}, \Sigma_{d-k})$, where $\Sigma_k$ and $\Sigma_{d-k}$ are defined as in Remark 5.2. By Corollary A.7, with probability at least $1 - \frac{0.02}{t}$, $s_k(Y^j) \in \Theta((\sqrt{m} + \sqrt{k})(\sqrt{s_k(\Sigma_k)})) = \Theta(\sqrt{m} + \sqrt{k}) > 0$. Therefore, the subspace spanned by $Y^j$ is the same as the subspace spanned by $\Sigma_k$. So, it suffices to look at the subspace spanned by $Y^j$.

Now, by Corollary A.7, we know that with probability at least $1 - \frac{0.02}{t}$, $\|X^j - Y^j\| = \|H\| \leq O((\sqrt{m} + \sqrt{d})\sqrt{s_1(\Sigma_{d-k})}) \leq O(\gamma(\sqrt{m} + \sqrt{d})\sqrt{s_k(\Sigma_k)}) \leq O(\gamma(\sqrt{m} + \sqrt{d}))$.

We wish to invoke Lemma A.3. Let $UDV^T$ be the SVD of $Y^j$, and let $\hat{U}\hat{D}\hat{V}^T$ be the SVD of $X^j$. Now, for a matrix $M$, let $\Pi_M$ denote the projection matrix of the subspace spanned by the columns

of $M$. Define quantities $a, b, z_{12}, z_{21}$ as follows.

$$
\begin{aligned}
a &= s_{\min}(U^T X^j V) \\
&= s_{\min}(U^T Y^j V + U^T H V) \\
&= s_{\min}(U^T Y^j V) \qquad\qquad \text{(Columns of } U \text{ are orthogonal to columns of } H) \\
&= s_k(Y^j) \\
&\in \Theta(\sqrt{m} + \sqrt{k}) \\
&\in \Theta(\sqrt{m}) \\
b &= \|U_\perp^T X^j V_\perp\| \\
&= \|U_\perp^T Y^j V_\perp + U_\perp^T H V_\perp\| \\
&= \|U_\perp^T H V_\perp\| \qquad\qquad \text{(Columns of } U_\perp \text{ are orthogonal to columns of } Y^j) \\
&\le \|H\| \\
&\le O(\gamma(\sqrt{m} + \sqrt{d})) \\
z_{12} &= \|\Pi_U H \Pi_{V_\perp}\| \\
&= 0 \\
z_{21} &= \|\Pi_{U_\perp} H \Pi_V\| \\
&= \|\Pi_{U_\perp} \Sigma_{d-k}^{1/2} (\Sigma_{d-k}^{-1/2} H) \Pi_V\|
\end{aligned}
$$

Now, in the above, $\Sigma_{d-k}^{-1/2} H \in \mathbb{R}^{d \times m}$, such that each of its entry is an independent sample from $\mathcal{N}(0, 1)$. Right-multiplying it by $\Pi_V$ makes it a matrix in a $k$-dimensional subspace of $\mathbb{R}^m$, such that each row is an independent vector from a spherical Gaussian. Using Corollary A.7, $\|\Sigma_{d-k}^{-1/2} H\| \le O(\sqrt{d} + \sqrt{k}) \le O(\sqrt{d})$ with probability at least $1 - \frac{0.01}{t}$. Also, $\|\Pi_{U_\perp} \Sigma_{d-k}^{1/2}\| \le O(\gamma \sqrt{s_k(\Sigma_k)}) \le O(\gamma)$. This gives us:

$$
z_{21} \le O(\gamma \sqrt{d}).
$$

Since $a^2 > 2b^2$, we get the following by Lemma A.3.

$$
\begin{aligned}
\|\text{Sin}(\Theta)(U, \hat{U})\| &\le \frac{a z_{21} + b z_{12}}{a^2 - b^2 - \min\{z_{12}^2, z_{21}^2\}} \\
&\le O\left(\frac{\gamma \sqrt{d}}{\sqrt{m}}\right)
\end{aligned}
$$

Therefore, using Lemma A.4, and applying the union bound over all $j$, we get the required result. $\qquad\square$

Let $\xi = O\left(\frac{\gamma \sqrt{d}}{\sqrt{m}}\right)$. We show that the projections of any reference point are close.

**Corollary D.2.** *Let $p_1, \dots, p_q$ be the reference points as defined in Algorithm 2, and let $\Pi$ and $\Pi_j$ (for $1 \le j \le t$) be projections matrices as defined in Lemma D.1. Then*

$$
\mathbb{P}\left[\forall i, j, \|(\Pi - \Pi_j) p_i\| \le O(\xi(\sqrt{k} + \sqrt{\ln(qt)}))\right] \ge 0.9.
$$

*Proof.* We know from Lemma D.1 that $\|\Pi - \Pi_j\| \le \xi$ for all $j$ with probability at least $0.95$. For $j \in [t]$, let $\widehat{\Pi}_j$ be the projection matrix for the union of the $j^{\text{th}}$ subspace and the subspace spanned by $\Sigma_k$. Lemma A.10 implies that with probability at least $0.95$, for all $i, j$, $\|\widehat{\Pi}_j p_i\| \le O(\sqrt{k} + \sqrt{\ln(qt)})$. Therefore,

$$
\|(\Pi - \Pi_j) p_i\| = \|(\Pi - \Pi_j)\widehat{\Pi}_j p_i\| \le \|\Pi - \Pi_j\| \cdot \|\widehat{\Pi}_j p_i\| \le O(\xi(\sqrt{k} + \sqrt{\ln(qt)})).
$$

Hence, the claim. $\qquad\square$

The above corollary shows that the projections of each reference point lie in a ball of radius $O(\xi\sqrt{k})$. Next, we show that for each reference point, all the projections of the point lie inside a histogram cell with high probability. For notational convenience, since each point in $Q$ is a concatenation of the projection of all reference points on a given subspace, for all $i, j$, we refer to $(0, \ldots, 0, Q^j_{(i-1)d+1}, \ldots, Q^j_{id}, 0, \ldots, 0) \in R^{qd}$ (where there are $(i-1)d$ zeroes behind $Q^j_{(i-1)d+1}$, and $(q-i)d$ zeroes after $Q^j_{id}$) as $p^j_i$.

**Lemma D.3.** *Let $\ell$ and $\lambda$ be the length of a histogram cell and the random offset respectively, as defined in Algorithm 2. For each $1 \le i \le q$, define the following event.*

$$E_i \equiv \exists \omega \in \Omega : \left| \omega \cap \{p^1_i, \ldots, p^t_i\} \right| = t$$

*Then $\mathbb{P}[E_i \cap \cdots \cap E_q] \ge 0.8$. Thus there exists $\omega \in \Omega$ that, such that all points in $Q$ lie within $\omega$.*

*Proof.* Let $r = O(\xi(\sqrt{k} + \sqrt{\ln(qt)}))$. This implies that $\ell = 20rq$. The random offset could also be viewed as moving along a diagonal of a cell by $\lambda \ell \sqrt{dq}$. We know that with probability at least $0.8$, for each $i$, all projections of reference point $p_i$ lie in a ball of radius $r$. Fix an $i \in [q]$. Then

$$\mathbb{P}\left[\overline{E_i}\right] \le \mathbb{P}\left[\frac{1}{20q} \ge \lambda \vee \lambda \ge \frac{19}{20q}\right] = \frac{1}{10q}.$$

Taking the union bound over all $q$ and the failure of the event in Corollary D.2, we get the first part of the claim. Since $p^j_i$'s are non-zero in disjoints sets of coordinates, the second part follows. $\qquad\square$

Now, we analyse the sample complexity due to the private algorithm, that is, DP-histograms.

**Lemma D.4.** *For each $1 \le i \le q$, let $\omega_i$ be the histogram cell as defined in Algorithm 2. If $t \ge O\left(\frac{\log(1/\delta)}{\varepsilon}\right)$, then $\mathbb{P}\left[\forall i, |\omega_i \cap \{p^1_i, \ldots, p^t_i\}| = t\right] \ge 0.75$.*

*Proof.* Lemma D.3 implies that with probability at least $0.8$, for each $i$, all projections of $p_i$ lie in a histogram cell, that is, all points of $Q$ lie in a histogram cell in $\Omega$. Because of the error bound in Lemma A.15 and our bound on $t$, we see at least $\frac{q}{2}$ points in that cell with probability at least $1 - 0.05$. Therefore, by taking the union bound, the proof is complete. $\qquad\square$

We finally show that the error of the projection matrix that is output by Algorithm 2 is small.

**Lemma D.5.** *Let $\widehat{\Pi}$ be the projection matrix as defined in Algorithm 2, and $n$ be the total number of samples. If*

$$\gamma^2 \in O\left(\frac{\varepsilon \alpha^2 n}{d^2 k^3 \ln(1/\delta)} \cdot \min\left\{\frac{1}{k}, \frac{1}{\ln(k \ln(1/\delta)/\varepsilon)}\right\}\right),$$

$n \ge O\left(\frac{k \log(1/\delta)}{\varepsilon} + \frac{\ln(1/\delta)\ln(\ln(1/\delta)/\varepsilon)}{\varepsilon}\right)$, and $q \ge O(k)$ the with probability at least $0.7$, $\|\widehat{\Pi} - \Pi\| \le \alpha$.

*Proof.* For each $i \in [q]$, let $p^*_i$ be the projection of $p_i$ on to the subspace spanned by $\Sigma_k$, $\widehat{p}_i$ be as defined in the algorithm, and $p^j_i$ be the projection of $p_i$ on to the subspace spanned by the $j^{\text{th}}$ subset of $X$. From Lemma D.4, we know that all $p^j_i$'s are contained in a histogram cell of length $\ell$. This implies that $\|p^j_i - \widehat{p}_i\| \le \ell\sqrt{dq}$. Since $p^j_i$'s and $p^*_i$ are contained in a ball of radius $\xi\sqrt{3d}$, it must be the case that $\|\widehat{p}_i - p^*_i\| \le 2\ell\sqrt{dq}$.

Now, let $P = (p^*_1, \ldots, p^*_q)$ and $\widehat{P} = (\widehat{p}_1, \ldots, \widehat{p}_q)$. Then by above, $\widehat{P} = P + E$, where $\|E\|_F \le 2\ell\sqrt{dq}$. Therefore, $\|E\| \le 2\ell\sqrt{dq}$. Let $E = E_P + E_{\overline{P}}$, where $E_P$ is the component of $E$ in the subspace spanned by $P$, and $E_{\overline{P}}$ be the orthogonal component. Let $P' = P + E_P$. We will be analysing $\widehat{P}$ with respect to $P'$.

Now, with probability at least $0.95$, $s_k(P) \in \Theta(\sqrt{k})$ due to our choice of $q$ and using Corollary A.7, and $s_{k+1}(P) = 0$. So, $s_{k+1}(P') = 0$ because $E_P$ is in the same subspace as $P$. Now, using Lemma A.2, we know that $s_k(P') \ge s_k(P) - \|E_P\| \ge \Omega(\sqrt{k}) > 0$. This means that $P'$ has rank $k$, so the subspaces spanned by $\Sigma_k$ and $P'$ are the same.

As before, we will try to bound the distance between the subspaces spanned by $P'$ and $\widehat{P}$. Note that using Lemma A.1, we know that $s_k(P') \leq s_k(P) + \|E_P\| \leq O(\sqrt{k})$.

We wish to invoke Lemma A.3 again. Let $UDV^T$ be the SVD of $P'$, and let $\hat{U}\hat{D}\hat{V}^T$ be the SVD of $\widehat{P}$. Now, for a matrix $M$, let $\Pi_M$ denote the projection matrix of the subspace spanned by the columns of $M$. Define quantities $a, b, z_{12}, z_{21}$ as follows.

$$
\begin{aligned}
a &= s_{\min}(U^T \widehat{P} V) \\
&= s_{\min}(U^T P' V + U^T E_{\overline{P}} V) \\
&= s_{\min}(U^T P' V) && \text{(Columns of } U \text{ are orthogonal to columns of } E_{\overline{P}}) \\
&= s_k(P') \\
&\in \Theta(\sqrt{k}) \\
b &= \|U_\perp^T \widehat{P} V_\perp\| \\
&= \|U_\perp^T P' V_\perp + U_\perp^T E_{\overline{P}} V_\perp\| \\
&= \|U_\perp^T E_{\overline{P}} V_\perp\| && \text{(Columns of } U_\perp \text{ are orthogonal to columns of } P') \\
&\leq \|E_{\overline{P}}\| \\
&\leq O(\ell\sqrt{d}q) \\
z_{12} &= \|\Pi_U E_{\overline{P}} \Pi_{V_\perp}\| \\
&= 0 \\
z_{21} &= \|\Pi_{U_\perp} E_{\overline{P}} \Pi_V\| \\
&\leq \|E_{\overline{P}}\| \\
&\leq O(\ell\sqrt{d}q)
\end{aligned}
$$

Using Lemma A.3, we get the following.

$$
\begin{aligned}
\|\text{Sin}(\Theta)(U,\hat{U})\| &\leq \frac{a z_{21} + b z_{12}}{a^2 - b^2 - \min\{z_{12}^2, z_{21}^2\}} \\
&\leq O\left(\ell\sqrt{dk}\right) \\
&\leq \alpha
\end{aligned}
$$

This completes our proof. $\qquad\square$

# E  Boosting

In this section, we discuss boosting of error guarantees of Algorithm 2. The approach we use is very similar to the well-known Median-of-Means method: we run the algorithm multiple times, and choose an output that is close to all other "good" outputs. We formalise this in Algorithm 3.

Now, we present the main result of this section.

**Theorem E.1.** *Let $\Sigma \in \mathbb{R}^{d \times d}$ be an arbitrary, symmetric, PSD matrix of rank $\geq k \in \{1, \ldots, d\}$, and let $0 < \gamma < 1$. Suppose $\Pi$ is the projection matrix corresponding to the subspace spanned by the vectors of $\Sigma_k$. Then given*

$$
\gamma^2 \in O\left(\frac{\varepsilon\alpha^2 n}{d^2 k^3 \ln(1/\delta)} \cdot \min\left\{\frac{1}{k}, \frac{1}{\ln(k\ln(1/\delta)/\varepsilon)}\right\}\right),
$$

*such that $\lambda_{k+1}(\Sigma) \leq \gamma^2 \lambda_k(\Sigma)$, for every $\varepsilon, \delta > 0$, and $0 < \alpha, \beta < 1$, there exists and $(\varepsilon, \delta)$-DP algorithm that takes*

$$
n \geq O\left(\frac{k\log(1/\delta)\log(1/\beta)}{\varepsilon} + \frac{\log(1/\delta)\log(\log(1/\delta)/\varepsilon)\log(1/\beta)}{\varepsilon}\right)
$$

*samples from $\mathcal{N}(\vec{0}, \Sigma)$, and outputs a projection matrix $\widehat{\Pi}$, such that $\|\Pi - \widehat{\Pi}\| \leq \alpha$ with probability at least $1 - \beta$.*

---

**Algorithm 3:** DP Approximate Subspace Estimator Boosted $\text{DPASEB}_{\varepsilon,\delta,\alpha,\beta,\gamma,k}(X)$

---

**Input:** Samples $X_1, \ldots, X_n \in \mathbb{R}^d$. Parameters $\varepsilon, \delta, \alpha, \beta, \gamma, k > 0$.
**Output:** Projection matrix $\widehat{\Pi} \in \mathbb{R}^{d \times d}$ of rank $k$.

Set parameters: $t \leftarrow C_3 \log(1/\beta) \qquad m \leftarrow \lfloor n/t \rfloor$

Split $X$ into $t$ datasets of size $m$: $X^1, \ldots, X^t$.

// Run DPASE $t$ times to get multiple projection matrices.
**For** $i \leftarrow 1, \ldots, t$
    $\widehat{\Pi}_i \leftarrow \text{DPASE}_{\varepsilon,\delta,\alpha,\gamma,k}(X^i)$
// Select a good subspace.
**For** $i \leftarrow 1, \ldots, t$
    $c_i \leftarrow 0$
    **For** $j \in [t] \setminus \{i\}$
        **If** $\|\widehat{\Pi}_i - \widehat{\Pi}_j\| \leq 2\alpha$
            $c_i \leftarrow c_i + 1$
    **If** $c_i \geq 0.6t - 1$
        **Return** $\widehat{\Pi}_i$.
// If there were not enough good subspaces, return $\perp$.
**Return** $\perp$.

---

*Proof.* Privacy holds trivially by Theorem 1.2.

We know by Theorem 1.2 that for each $i$, with probability at least 0.7, $\|\widehat{\Pi}_i - \Pi\| \leq \alpha$. This means that by Lemma A.9, with probability at least $1 - \beta$, at least $0.6t$ of all the computed projection matrices are accurate.

This means that there has to be at least one projection matrix that is close to $0.6t - 1 > 0.5t$ of these accurate projection matrices. So, the algorithm cannot return $\perp$.

Now, we want to argue that the returned projection matrix is accurate, too. Any projection matrix that is close to at least $0.6t - 1$ projection matrices must be close to at least one accurate projection matrix (by pigeonhole principle). Therefore, by triangle inequality, it will be close to the true subspace. Therefore, the returned projection matrix is also accurate. $\qquad\square$