# OpenReview forum: "Privately Learning Subspaces"
_NeurIPS.cc/2021/Conference — NeurIPS 2021 Poster_

### Official Review · Reviewer_Rxwy · 2021-07-05

**Rating:** 7
**Confidence:** 4

**Summary:**

The paper looks at the problem of privately learning a subspace. They consider two different settings and give algorithms for each setting.

In the first setting, they assume that almost all the points lie in a fixed subspace of dimension $k$. They also make a non-degeneracy assumption which states that no subspace of dimension $k-1$ contains too many of the points. Under these assumptions, they give an algorithm which returns the subspace exactly and with probability 1. Their algorithm has the optimal sample complexity. However, it is not efficient except for constant $k$. One interesting aspect of the result is that it has no dependence on the ambient dimension.

In the second setting, they make a distributional assumption. They assume that the points come from a mean-zero Gaussian with some unknown covariance matrix. The covariance matrix is assumed to have a large gap between the $k$th largest and the $(k+1)$th largest eigenvalue. Under these assumptions, the authors given an algorithm which returns an approximation to the subspace. Here, approximation is measured as the operator norm in the difference between the respective projection matrices.

**Limitations And Societal Impact:**

Yes

**Main Review:**

**Originality.**
The paper considers the problem of privately learning a subspace. This is a basic and fundamental problem which has not been previously studied in the literature although there have been quite a bit of work in related areas such as privately PCA. In terms of techniques, the ideas used in the exact case are a somewhat simple application of the GAP-MAX strategy with a nice score function. The approximate setting makes use of many more sophisticated techniques in a nice way (in particular, to prove accuracy) such as matrix perturbation and matrix concentration results. Their idea of using reference points as a way to aggregate candidate projection matrices is quite clever and nice.

**Quality.**
The submission appears to be technically sound as far as I can tell. All the claims either have proofs in the main part of the paper or the appendix. However, I have not verified or read over the proofs in the appendix.

**Clarity.**
Overall, the writing quality of the paper is quite good. The authors do a good job in explaining their contributions and the technical ideas. In the exact setting, I was able to easily follow the details and understand why the algorithm works. The approximate case is much more technical but the authors clearly explain provide some high-level intuition.

**Significance.**
I believe this is a good starting point for the problem of privately learning subspaces. The present work may be of interest to researchers designing algorithms for high-dimensional datasets which may possess some lower-dimensional structure. In these settings, the present work may be of value and may help reduce the dimensionality in a private manner.

**Other comments.**
One minor downside I have of the paper is that in the exact case, the algorithm is not very efficient (except for constant $k$). I wonder if the authors have thought a bit about how to design faster algorithms.
Another minor downside is that the required gap for the approximate case seems to be quite large and works only if it happened to be the case where the gap between the kth and (k+1)th eigenvalue is large.

I have one additional question for the authors: In the exact case, how is the subspace represented? For example, one can use any k linearly independent vectors to represent the vectors but this is obviously not private. So I am wondering if the authors could elaborate a bit more on this aspect.

Lemma 4.3: Should be $u(x,s') \leq \ell$. This is what is proved and is what is used in Lemma 4.4.

In the proof of Lemma 4.3, first paragraph, it might be good to mention $|x \cap s_*| \geq n-\ell$ by assumption (i) as well, just to increase clarity.

**Update** The authors were very clear in answering my questions and I like the paper. So have increased my score.


**Time Spent Reviewing:**

4

---

> ### Author Response · Authors · 2021-08-09
> **Response to Reviewer Rxwy**
>
> Thank you for your detailed review and for your low-level comments, which we will fix/add.
>
> Re. efficiency: Our exact algorithm (Algorithm 1) is inefficient, but our approximate algorithm (Algorithm 2) is efficient. We believe that the runtime in the exact case could be improved; see our response to all reviewers.
>
> Re. eigengap: The required eigengap is indeed large. We believe this can be improved somewhat, but some eigengap assumption is inherently necessary. Even without privacy, we would require an eigengap that grows with the ambient dimension $d$.
>
> Re. representation of a subspace: This is an excellent question. In the exact case, we can simply represent the subspaces by the orthogonal projection matrix onto that space. However, finding a representation of the subspace that doesn’t itself violate privacy is a major challenge in the approximate case and this is what led us to use projections of random reference points.

---

> > ### Comment · Reviewer_Rxwy · 2021-08-30
> > **thanks**
> >
> > Thanks for answering my questions clearly. I like the results in the paper and have increased my score.

---

### Official Review · Reviewer_P1nL · 2021-07-12

**Rating:** 6
**Confidence:** 3

**Summary:**

The paper studies the problem of learning a low-dimensional subspace from high-dimensional data in a differentially private manner.  The emphasis is on getting sample complexity that only depends on the dimension of the subspace but not the dimension of the original space. Such a guarantee is not achieved by previous techniques such as private PCA.

Formally, the paper studies two settings: the exact setting and the approximate setting.  In the exact setting, they assume that they are given $n$ points and almost all of them are contained in some $k$-dimensional subspace.  It is also necessary to make a non-degeneracy assumption that no $k-1$-dimensional subspace contains too many of the points.  In the approximate setting, they assume that all of the points are drawn from a Gaussian that is close to $k$-dimensional (i.e. its covariance matrix has $k$ large eigenvalues and $d-k$ much smaller eigenvalues).  In the first case, they obtain a sample complexity of $O(k + \log (1/\delta)/\epsilon)$  and in the second setting they obtain a sample complexity of  $O(k \log(1/\delta)/\epsilon)$ for achieving $(\epsilon, \delta)$ differential privacy.  Crucially, these are linear in $k$ and independent of $d$ and also their privacy guarantee is in the worst case, where any of the sample points may be altered arbitrarily.  It is also worth noting that for the approximate setting, they do not really use the Gaussian distribution assumption very strongly and only need that the sample covariances approximate the true covariances.

**Main Review:**

Overall, the contributions are solid and the concept of getting sample complexity depending only on the `````"true" dimensionality of the data rather than the full dimensionality of the space is nice.  The paper is reasonably well-written and understandable.

 I do feel that the algorithms presented are somewhat unsatisfying because at their core they involve some type of brute-force search over all possible subspaces.  Thus, my main question to the authors is: do you think the algorithms presented in the paper can be made computationally efficient or would this require a significantly different approach?


After Author Response: Thanks for the clarifications!  My opinion remains that this paper is a solid, but not quite spectacular contribution.

**Time Spent Reviewing:**

2

---

> ### Author Response · Authors · 2021-08-09
> **Response to Reviewer P1nL**
>
> Thank you for your thorough review of our paper.
>
> The exact algorithm (Algorithm 1) enumerates O(n^k) subspaces, which is inefficient, but the approximate algorithm (Algorithm 2) is significantly more efficient as well as being more generally applicable. We believe that Algorithm 1 could be made more efficient by running over a random subset of the subspaces; please see our response to all reviewers.

---

### Official Review · Reviewer_YUHG · 2021-07-16

**Rating:** 8
**Confidence:** 4

**Summary:**

This paper studies the following problem: given d-dimensional data living in an (unknown) linear subspace of dimensionality k<d, learn the subspace under the constraint of differential privacy (DP).

The authors Provide the first DP method(s) for learning a low dimensional subspace that the data lives in, where the sample complexity of their algorithms can scale with k instead of d. Their algorithms can serve as an important private pre-processing step that can reduce the sample complexity of further estimation tasks.

The authors consider two different settings. In the first “exact case”, the authors assume the data lies in a subspace s* of dimensionality k < d. For technical reasons, the authors also make the assumption that any strict subspace of s* contains at most l points. Their final sample complexity bounds scale with l instead of d (this is good when l is small, aka ``nice’’) and they present an (eps,delta)-DP algorithm that outputs s* with probability 1. Finally they give a matching lower bound for this setting proving their result is sharp.

The result in the exact setting can be quite restrictive if the dataset is slightly perturbed (e.g. from measurement noise). This motivates the second “approximate case” in which the authors assume the data is generated from some 0 mean d-dimensional Gaussian. They assume a multiplicative ``eigenvalue gap’’ between the top k and bottom d-k eigenvalues and present an algorithm that takes advantage of the celebrated sub-sample and aggregate algorithm and stability based histogram. Their algorithm outputs a projection matrix that (with high probability) is close in operator norm to the “true” projection matrix (corresponding to the subspace spanned by the top k eigenvectors of the covariance of the Gaussian). Here, $\Pi$ is precisely what we want to recover since for further data analysis tasks, one could project to the low dimensional subspace and work in this subspace for future tasks. Finally, the sample complexity of this second algorithm grows polynomially with k instead of d.


**Limitations And Societal Impact:**

I don't see direct societal impact of this paper (aside from the benefits of studying privacy)

**Main Review:**

This is solid work that takes advantage of classic tools such as stability based histograms and the GAP-MAX algorithm in new and clever ways. The arguments (to prove the privacy and accuracy) are sound. The paper is well written and includes some high-level discussions.

The first algorithm presented in the “exact case” is very simple and also has the added benefit of being sample optimal and correct with probability 1. Unfortunately, the running time of this algorithm is exponential in the size of the dataset and the assumptions on the dataset don’t seem to be very realistic, making it unlikely to see practical application. Nevertheless this is an important result from a theoretical point of view.

The second algorithm handles the approximate case but makes a distributional assumption (Gaussian) and an eigengap (which is necessary even in the non DP case). It would have been nice if the authors have explicitly stated the running time of this algorithm.

It appears that the algorithm requires the knowledge of k, the underlying dimension of the low dimensional subspace. This seems to be unrealistic in practical applications. It would be nice if the authors comment on this.

Overall, the paper addresses a fundamental problem and should be attractive to both theoreticians and practitioners. More generally, the paper betters our understanding of how to exploit the structure of data to circumvent lower bounds for private estimation/learning tasks in high-dimensions.

Small typo: In line 296, The inequality $ \geq n -3l -4\log(1/\delta)/\varepsilon -1$ should end with -2 instead of -1 I believe.

**Time Spent Reviewing:**

5

---

> ### Author Response · Authors · 2021-08-09
> **Response to Reviewer YUHG**
>
> Thank you for your time and comments and for spotting a minor error in our analysis. We are pleased that you appreciate our results.
>
> Re. runtime: We will add statements about the runtime; please see our response to all reviewers. Thank you for pointing out this omission
>
> Re. knowing the parameter k = dimension of the true subspace: For the exact algorithm, we do not need to know k -- i.e. the algorithm could simply be run over all subspaces, or, if we have some upper bound k <= kmax, we can run it on all subspaces of dimension <=kmax. For the approximate algorithm, the parameter is more critical. However, there are generic tools available that would allow us to automatically set this parameter at a modest cost in privacy  [see, e.g., Liu & Talwar. Private Selection from Private Candidates. https://arxiv.org/abs/1811.07971 ]. It should be possible to select k without resorting to a heavy hammer like this; we will need to think more about a direct algorithm for selecting k.
>
> We will add a discussion about the parameter k in the next version of our paper. Thank you for the suggestion.

---

### Official Review · Reviewer_YS3D · 2021-07-17

**Rating:** 5
**Confidence:** 3

**Summary:**

This paper studies the problem of learning low-dimensional structure from the data subject to differential privacy. The paper considers the setting where the data lies in or very close to a linear subspace of dimension (k<<d). For the exact case where all the data lies in a linear subspace, the proposed algorithm DPESE, which applies the GAP-MAX, can recover the subspace exactly when there are enough samples. For the approximate case, they assume that the data comes from a Gaussian distribution where the covariance matrix has a certain eigenvalue gap. This paper proposes the DPASE algorithm that is based on the subsample-and-aggregate algorithm. In both cases, the sample complexity the dimension-independent.

**Limitations And Societal Impact:**

The paper has a discussion on the limitations of the paper. I don't see the paper has any potential negative societal impact.

**Main Review:**

The paper is well-written, and I found the analysis and algorithm easy to follow. This paper studies learning a linear subspace in the setting that the data lies in or is very close to the subspace. This problem is useful for dimension reduction and thus prevents the curse of dimensionality in private data analysis. It has some nice ideas. However, the technical contribution is not strong enough. The two algorithms are largely based on existing tools. It's nontrivial, but it's far from challenging to develop. The theoretical analysis is also quite standard, especially given that the assumptions are very strong. One question I have is how efficient to compute the scores in Algorithm 1. It seems you have to enumerate all the subspaces of a particular space.


**Time Spent Reviewing:**

5

---

> ### Author Response · Authors · 2021-08-09
> **Response to Reviewer YS3D**
>
> Thank you for your time and for your comments.
>
> The first algorithm for the exact case is indeed fairly standard and requires strong assumptions to work. We relax those assumptions in the second algorithm for the approximate case. We emphasize that the assumptions we make in the approximate case are relatively minimal, as these are the kinds of assumptions that need to be made even in the non-private version of the problem. Please, see our response to all reviewers.
>
> Regarding the strength of our technical contribution: This is, of course, subjective. From our perspective, our second algorithm for the approximate case was, at least for us, nontrivial to design and analyze. The idea of using random reference points is, to the best of our knowledge, novel and was somewhat of a breakthrough for us. We also needed to import tools from matrix perturbation theory and matrix concentration to give good accuracy bounds for our algorithm.
>
> Regarding the runtime: The first algorithm for the exact case requires enumerating O(n^k) subspaces. We believe this could be made more efficient by instead running on a randomly chosen subset of those subspaces, but we did not develop this idea because our second algorithm for the approximate case is more efficient anyway.

---

### Author Response · Authors · 2021-08-09
**Response to all reviewers**

We thank all of the reviewers for their time and helpful comments. They raise several interesting questions, which we believe demonstrates how this is an interesting topic for further research.

We respond to each reviewer individually, but address some common comments here.

**Assumptions.**

Even without privacy, the problem of learning a subspace from data requires some assumptions on that data. The assumptions we make for our private algorithms are qualitatively the same as would be necessary in the non-private setting. Although, as usual, they are quantitatively slightly stronger in the private setting.

In the exact case, we assume that the true subspace contains “many” points and any other subspace contains “few” points, which is exactly what we would need to assume non-privately. The only difference is that the thresholds for quantifying “many” and “few” now depend on the privacy parameters.

In the approximate case, we assume that the data has some distribution, namely Gausian, and that there is a large eigengap in the covariance matrix. The distributional assumption is required to ensure that our estimates of the covariance matrix are close to the true covariance matrix with high probability. This is exactly the same as would be required in the non-private setting. We can substitute in a different distributional assumption (such as, distributions with bounded $m$-th moment), as long as there is a comparable matrix concentration bound that we could use.

The eigengap assumption is necessary to ensure that the true subspace “stands out” enough for us to be able to accurately estimate it. This is also necessary without privacy; Cai and Zhang [CZ16] show that an eigengap with a polynomial dependence on the dimension is necessary in the setting we consider, even without privacy.

**Efficiency.**

The exact algorithm (Algorithm 1) enumerates $O(n^k)$ subspaces, which is inefficient, but the approximate algorithm (Algorithm 2) is based on the sample-and-aggregate framework and is significantly more efficient.

The dominant step in the runtime of the approximate algorithm is computing the covariance matrix and its eigendecomposition for each of the subsamples. This can be upper bounded by $\tilde{O}(d^4 n)$ operations; a more precise bound is $\tilde{O}(t \cdot d^{\omega+1} + n d^2)$ operations, where $t=O(\log(1/\delta)/\varepsilon)$ is the number of subsamples and $\omega<2.376$ is the matrix multiplication exponent. (Here we omit the dependence on the accuracy parameters and bit complexity of the operations.)

We believe that Algorithm 1 could be made more efficient by running over a subset of the subspaces, rather than all of them. Specifically, we could select a set of input points at random and consider the subspace spanned by these points. If we repeat this many times independently, then with high probability we will consider all of the “good” subspaces -- i.e. those containing many points. Thus running the gap-max algorithm over this random subset of subspaces would still be DP. However, we didn’t analyze this approach because we already have an efficient algorithm in Algorithm 2.

---

### Decision · Program_Chairs · 2021-09-27

**Decision:**

Accept (Poster)

**Comment:**

The paper studies the problem of differentially private subspace learning. Under an assumption on low-dimensional nature of data, the authors present algorithms that enjoy a sample complexity scaling with the dimension of the subspace and not with the ambient dimension, thereby improving on existing results. Overall, a good paper.